# INFERENCE-TIME SEARCH USING SIDE INFORMATION FOR DIFFUSION-BASED IMAGE RECONSTRUCTION

## ABSTRACT

Diffusion models have emerged as powerful priors for solving inverse problems. However, existing approaches typically overlook side information that could significantly improve reconstruction quality, especially in severely ill-posed settings. In this work, we propose a novel inference-time search algorithm that guides the sampling process using the side information in a manner that balances exploration and exploitation. This enables more accurate and reliable reconstructions, providing an alternative to the gradient-based guidance that is prone to reward-hacking artifacts. Our approach can be seamlessly integrated into a wide range of existing diffusion-based image reconstruction pipelines. Through extensive experiments on a number of inverse problems, such as box inpainting, super-resolution, and various deblurring tasks including motion, Gaussian, nonlinear, and blind deblurring, we show that our approach consistently improves the qualitative and quantitative performance of diffusion-based image reconstruction algorithms. We also show the superior performance of our approach with respect to other baselines, including reward gradient-based guidance algorithms. Code is available here.

## 1 INTRODUCTION

Diffusion models (Ho et al., 2020; Song et al., 2021b) have demonstrated remarkable success in generative tasks across various fields like text-to-image synthesis (Rombach et al., 2022), protein sequence (Wu et al., 2024), video (Ho et al., 2022), audio (Kong et al., 2021), and language modeling (Austin et al., 2021; Sahoo et al., 2024). Aside from generation, these models have also shown great promise in *solving inverse problems*, where the goal is to reconstruct an image from partial or noisy observations (Chung et al., 2023b; Song et al., 2023a; Rout et al., 2023; Song et al., 2024; He et al., 2024; Ye et al., 2024; Zhang et al., 2024). Inverse problems differ fundamentally from standard generative tasks (e.g., text-to-image synthesis, personalized editing, style transfer): whereas those tasks are often judged subjectively, inverse problems have a precise objective, to recover a specific ground-truth signal from incomplete measurements. Consequently, fidelity to the ground truth and rigorous quantitative evaluation are critical.

When the observation is heavily degraded, the inverse problem becomes highly ill-posed as many distinct signals can explain the data almost equally well. In this regime, unconstrained posterior sampling rarely recovers the ground truth; it tends instead to produce diverse yet semantically inconsistent reconstructions. A practical solution is to incorporate **side information**, auxiliary measurements correlated with the target signal, to constrain the solution space and steer the reconstruction toward faithful outcomes. This idea is well established in the classical signal processing literature, where certain structural or encoded properties are used to guide the iterative algorithms that solve the inverse problem (Jones, 2009; Chun et al., 2012; Oymak et al., 2013; Ehrhardt et al., 2014; Mota et al., 2017; Hyder et al., 2019). In medical imaging, leveraging complementary measurements or modalities, such as multiple MRI contrasts, multimodal microscopy, or RGB guidance for NIR imaging, has been shown to substantially improve quality (Atalık et al., 2025; Tsiligianni & Deligiannis, 2019).

While the existing works on diffusion-based solvers have made significant progress on measurement-only inverse problems, they largely sidestep the harder and increasingly common setting where we must also exploit side information (e.g., a reference photograph of the same person, a text description, or features from another modality). A key obstacle is the challenge of learning the conditional distribution $p_{X|Y,S}$, where $X$ denotes the target image, $Y$ denotes the noisy measurement, and $S$ denotes the side information. While some recent works (Kim et al., 2025a; Chung et al., 2025) address the limited setting of textual side information, these approaches typically train a diffusion

Figure 1: Illustration of the performance of our inference-time search algorithm for using side information in solving inverse problems, compared with the DPS algorithm (Chung et al., 2023b).

model to take a specific side-info modality as input; this demands large paired datasets and expensive training, ties the solver to a single conditioning format, and is impractical when the test-time side information differs from what the model was trained on. This motivates us to address the following question:

*How can we leverage a pre-trained (unconditional) diffusion prior to solve inverse problems with side information at inference time, without any retraining, so that the method is modality-agnostic and can use text, images, or features depending on the end-use applications?* We provide constructive solutions to these questions in our work. Our main contributions are the following.

- **Modeling:** We introduce a general modeling approach that incorporates arbitrary side information via an auxiliary reward, characterizing $p_{X|S}$ as a reward-tilted version of the pre-trained diffusion prior. This abstraction cleanly decouples the measurement model from the side information, is modality-agnostic (text, image, features), and requires no retraining. We use this modeling with tractable approximations and appropriate error bounds for computing the conditional score functions that are needed for sampling from the pre-trained diffusion models.

- **Algorithm:** Motivated by recent successes of inference-time search in LLMs (Snell et al., 2025; Setlur et al., 2025; Liu et al., 2025), we propose a compute-aware, training-free inference-time search framework that can leverage the side information to solve inverse problems. We instantiate this framework by proposing two specific search algorithms: $(i)$ Greedy Search (GS), a strategy that resamples greedily at each step, and $(ii)$ Recursive Fork-Join Search (RFJS), which balances exploration and exploitation through a group-based sampling at each step. The framework operates as a plug-in on top of any standard inverse-problem solvers and supports black-box, non-differentiable rewards. To our knowledge, our work is the first to propose inference-time search with side information for diffusion-based inverse problems.

- **Experiments:** We provide extensive experimental evaluations of our proposed approach across linear and nonlinear problems (e.g., box inpainting, super-resolution, motion/Gaussian/nonlinear deblurring) and side-information types (images and text), and demonstrate that our approach outperforms multiple relevant baseline algorithms.

## 2 RELATED WORK

**Inverse problems with diffusion priors:** Diffusion models (Dhariwal & Nichol, 2021; Ho et al., 2020; Song & Ermon, 2019; Sohl-Dickstein et al., 2015; Song & Ermon, 2020; Song et al., 2021a) are powerful generative models that sample from data distributions by iteratively denoising random noise. Several works adapt diffusion priors to inverse problems via likelihood score approximations. Diffusion Posterior Sampling (DPS) (Chung et al., 2023b) is a foundational method for solving inverse problems in a principled way. Its key idea is to approximate the expected conditional likelihood by evaluating the likelihood at the conditional mean, effectively pushing the expectation through the nonlinear function (Sec. 3.1). ΠGDM (Song et al., 2023a) solves linear inverse problems using a better

approximation than DPS; MPGD (He et al., 2024) avoids this cost by enforcing data consistency in image space; MCG (Chung et al., 2022) constrains reconstructions via manifold projections; DDRM (Kawar et al., 2022) operates in spectral space; and DAPS (Zhang et al., 2024) decouples diffusion steps. Latent diffusion priors are also used: PSLD (Rout et al., 2023) adds consistency terms, ReSample (Song et al., 2024) solves per-step optimization problems, and Chung et al. (2024) tunes prompts for efficiency. None of these methods, however, leverage side information.

**Inverse problems with side information:** Many works in signal processing (Mota et al., 2017; Oymak et al., 2013; Jones, 2009; Chun et al., 2012; Ehrhardt et al., 2014; Hyder et al., 2019) integrate structural correlations from auxiliary signals, often via designing appropriate optimization algorithms. In MRI, LeSITA (Tsiligianni & Deligiannis, 2019) learns coupled sparse representations, and TGVN (Atalık et al., 2025) constrains ambiguous subspaces with additional contrasts using learned unrolled networks. Diffusion-based approaches include training with joint priors across modalities (Levac et al., 2023; Efimov et al., 2025), metadata conditioning (Chung et al., 2025), and text-guided regularization (Kim et al., 2025a). Most approaches, however, are training-based or bound to one modality of side information associated with the trained conditional diffusion model.

**Reward-gradient guidance:** LGD (Song et al., 2023b) refines DPS via Monte Carlo estimates, while UGD (Bansal et al., 2024), FreeDoM (Yu et al., 2023), and RB-Modulation (Rout et al., 2025) propose to guide the diffusion with a gradient of the reward function. In addition to being gradient-based approaches, they are typically used for semantic generation tasks rather than inverse problems.

**SMC methods:** Sequential Monte Carlo approaches (Cardoso et al., 2024; Dou & Song, 2024; Wu et al., 2023) generate and resample particles under tilted distributions, offering gradient-free alternatives but limited performance at small $N$. DAS (Kim et al., 2025b) combines resampling with gradients for text-to-image tasks. These methods rely only on the measurement to guide the unconditional sampler and do not exploit side information.

**Inference-time search:** Reward-guided inference-time search has advanced LLM reasoning using Process Advantage Verifiers (PAVs) (Setlur et al., 2025), compute-optimal scheduling (Snell et al., 2025), and reward-guided small models (Liu et al., 2025). Some recent works (Singhal et al., 2025; Li et al., 2025) apply reward-based search in diffusion for text-to-image/protein generation, but do not consider side information or inverse problems.

## 3 PRELIMINARIES AND PROBLEM FORMULATION

### 3.1 PRELIMINARIES

**Diffusion models:** Diffusion models (Ho et al., 2020; Song et al., 2021b) are powerful generative models that enable sampling from an (unknown) distribution through an iterative process. Diffusion models comprise a forward diffusion process and a reverse denoising process. During the forward process, a clean sample from the distribution $p_{\text{data}}$ is progressively corrupted by the addition of Gaussian noise at each timestep, transforming the data distribution into pure noise. Conversely, the reverse process trains a denoising neural network to iteratively remove this introduced noise, enabling the reconstruction of samples from the initial data distribution. The forward process is represented by the stochastic differential equation (SDE), $d\mathbf{x}_t = f(\mathbf{x}_t, t)dt + g(t)d\mathbf{w}_t, \ \forall t \in [0, T]$, where $x_0$ is sampled from $p_{\text{data}}$ and $\mathbf{w}_t$ is a Wiener process. Common choices for $f, g$ are $f(\mathbf{x}_t, t) = -(\beta(t)/2)\mathbf{x}_t$ and $g(t) = \sqrt{\beta(t)}$ for some non-negative monotonic increasing function $\beta(\cdot)$ over $[0, T]$. The corresponding reverse process of this SDE is described by (Anderson, 1982; Song et al., 2021b) $d\mathbf{x}_t = (f(\mathbf{x}_t, t) - g^2(t)\nabla_{\mathbf{x}_t} \log p_t(\mathbf{x}_t)) dt + d\mathbf{w}_t, \ \forall t \in [T, 0]$, where $p_t$ denotes the marginal probability distribution of $\mathbf{x}_t$, $\mathbf{x}_T$ is sampled according to a standard Gaussian distribution, and $\nabla_{\mathbf{x}_t} \log p_t(\mathbf{x}_t)$ represents the *score function*. Since the marginal distribution $p_t$ is unknown, the score function is approximated by a neural network $\mathcal{D}_{\boldsymbol{\theta}}(\mathbf{x}_t, t)$ via the minimization of a score-matching objective. In practical implementations, the SDE is discretized into $T$ steps, and we define $\alpha_t \triangleq \prod_{s=1}^{t}(1 - \beta_s)$.

**Solving inverse problems using diffusion models:** An inverse problem consists of recovering an unknown signal $\mathbf{x}_0$ from noisy, partial observations $\mathbf{y} = \mathbf{A}(\mathbf{x}_0) + \sigma_y \mathbf{z}$, where $\mathbf{A}$ is the measurement model, $\sigma_y$ is the observation noise level, and $\mathbf{z}$ is typically a Gaussian noise. Often, $\mathbf{A}$ is non-injective, i.e., multiple signals $\mathbf{x}_0$ can produce the same measurement $\mathbf{y}$. A standard approach for estimating $\mathbf{x}_0$ is via the Bayesian framework, assuming a prior distribution $p_0$ over the signal $\mathbf{x}_0$, and sampling from the posterior distribution $\mathbf{x}_0 \sim p_{0|Y}(\cdot \mid \mathbf{y})$. Though $p_{0|Y}(\cdot \mid \mathbf{y})$ is not

known, this sampling can be achieved by running the backward SDE with replacing the original score function with the conditional score function $\nabla_{\mathbf{x}_t} \log p_{t|Y}(\mathbf{x}_t \mid \mathbf{y})$. Using Bayes' theorem, $\nabla_{\mathbf{x}_t} \log p_{t|Y}(\mathbf{x}_t \mid \mathbf{y}) = \nabla_{\mathbf{x}_t} \log p_t(\mathbf{x}_t) + \nabla_{\mathbf{x}_t} \log p_{Y|t}(\mathbf{y} \mid \mathbf{x}_t)$. While the score function network $\mathcal{D}_{\boldsymbol{\theta}}$ of the pre-trained diffusion model can be used to approximate the first term, approximating the second term is significantly more challenging, and numerous approaches (Daras et al., 2024) have been proposed to tackle this challenge. In particular, Diffusion Posterior Sampling (DPS) (Chung et al., 2023b) proposes a simple approach to approximate $p_{Y|t}$ as $p_{Y|t}(\mathbf{y} \mid \mathbf{x}_t) = \mathbb{E}_{\mathbf{x}_0 \sim p_{0|t}(\cdot|\mathbf{x}_t)}[p_{Y|0}(\mathbf{y} \mid \mathbf{x}_0)] \approx p_{Y|0}(\mathbf{y} \mid \mathbb{E}_{\mathbf{x}_0 \sim p_{0|t}(\cdot|\mathbf{x}_t)}[\mathbf{x}_0])$, by pushing the expectation inside the nonlinear $p_{Y|0}(\mathbf{y} \mid \cdot)$. The remaining challenge is to compute the conditional mean $\mathbb{E}_{\mathbf{x}_0 \sim p_{0|t}(\cdot|\mathbf{x}_t)}[\mathbf{x}_0] \triangleq \hat{\mathbf{x}}_{0|t}(\mathbf{x}_t)$, which is typically tackled by using Tweedie's formula (Efron, 2011), leveraging the fact that $\mathbf{x}_t$ given $\mathbf{x}_0$ is Gaussian. This results in the estimate

$$\hat{\mathbf{x}}_{0|t}(\mathbf{x}_t) = (1/\sqrt{\alpha_t})(\mathbf{x}_t + (1 - \alpha_t)\nabla_{\mathbf{x}_t} \log p_t(\mathbf{x}_t)) \approx (1/\sqrt{\alpha_t})(\mathbf{x}_t + (1 - \alpha_t)\mathcal{D}_{\boldsymbol{\theta}}(\mathbf{x}_t, t)). \quad (1)$$

### 3.2 PROBLEM FORMULATION: SOLVING INVERSE PROBLEMS WITH SIDE INFORMATION

In many applications, the observation $\mathbf{y}$ alone is insufficient to identify the latent signal $\mathbf{x}_0$; auxiliary side information $\mathbf{s}$ (e.g., a reference image, identity/text embedding, or physics-derived features) can dramatically reduce ambiguity. Formally, when side information $\mathbf{s}$ is available, *the goal is to sample from the target conditional distribution* $p_{0|Y,S}(\cdot \mid \mathbf{y}, \mathbf{s})$. A seemingly direct route is to *train a conditional diffusion model* that accepts $\mathbf{s}$ as input, learn the conditional score function $\nabla_{\mathbf{x}_t} \log p_{t|S}(\mathbf{x}_t \mid \mathbf{s})$, and then approximate the full conditional score $\nabla_{\mathbf{x}_t} \log p_{t|Y,S}(\mathbf{x}_t \mid \mathbf{y}, \mathbf{s}) = \nabla_{\mathbf{x}_t} \log p_{t|S}(\mathbf{x}_t \mid \mathbf{s}) + \nabla_{\mathbf{x}_t} \log p_{Y|t,S}(\mathbf{y} \mid \mathbf{x}_t, \mathbf{s})$ through a DPS-style method for the second term, to run the backward SDE. However, this training-based approach is often impractical: it demands large paired datasets $(\mathbf{x}_0, \mathbf{s})$, which are expensive or impossible to curate; it locks the solver to the training modality of $\mathbf{s}$ (a text-conditioned prior cannot natively exploit an image or spectral feature at test time); and general multi-modal conditioning requires prohibitive data and compute. These constraints motivate a training-free alternative that reuses strong unconditional diffusion priors and uses $\mathbf{s}$ only at inference, preserving modality-agnosticism and avoiding costly data collection.

Designing such a training-free method is technically challenging. First, DPS-style derivations rely on tractable likelihoods (e.g., Gaussian $p_{Y|0}$), whereas realistic $p_{S|0}$ are often non-Gaussian implicitly, complicating conditional-score construction. Second, even for measurement-only guidance, computing the conditional score used in the DPS-style algorithms requires back-propagating through the denoiser at every step. Naively extending to side information forces second-order/Hessian terms through the diffusion network. Third, purely gradient-guided diffusion is brittle: it struggles with non-differentiable or black-box rewards, amplifies early-step errors, and can drift off the data manifold. Inference-time search approaches, which have shown remarkable performance improvement in LLMs (Setlur et al., 2025; Liu et al., 2025; Snell et al., 2025) and text-conditioned diffusion models (Singhal et al., 2025; Kim et al., 2025b), but have not yet been used for solving the inverse problems, offer a promising path to overcome these challenges. In this context, we address the following questions:

($i$) *Modeling:* How can we realize $p_{0|Y,S}$ at inference time, without any retraining, by constructing a surrogate objective that is valid across diverse side-information modalities? ($ii$) *Algorithm*: How can we design a plug-and-play inference-time search module that is modality-agnostic, compute-aware, and capable of making global corrections (beyond local gradient steps)?

## 4 MODELING AND ALGORITHM

### 4.1 MODELING SIDE INFORMATION USING REWARD FUNCTION

Given a side-information signal $\mathbf{s}$ corresponding to an unknown $\mathbf{x}_0$, and two candiate reconstructions, $\mathbf{x}_0^1$ and $\mathbf{x}_0^2$, a principled way to decide which reconstruction is more truthful is to compare the (unknown) conditional probabilities $p_{0|S}(\mathbf{x}_0^1 \mid \mathbf{s})$ and $p_{0|S}(\mathbf{x}_0^2 \mid \mathbf{s})$. Directly estimating $p_{0|S}$ is intractable in our setting: it is typically non-Gaussian, multi-modal, and depends on the data domain and modality of $\mathbf{s}$. We therefore introduce a reward function $r : \mathbb{R}^d \times \mathcal{S} \to \mathbb{R}$ that orders reconstructions given $\mathbf{s}$: if $r(\mathbf{x}_0^1, \mathbf{s}) > r(\mathbf{x}_0^2, \mathbf{s})$, then $\mathbf{x}_0^1$ is deemed more compatible with $\mathbf{x}_0$ than $\mathbf{x}_0^2$. This abstraction aligns with many real-world applications (as shown in our experiments): when $\mathbf{s}$ is a text description of the target image $\mathbf{x}_0$, we can use a pre-trained text-image model to score text-image alignment. When $\mathbf{s}$ is a reference image of the same entity (e.g., the same person under different poses/lighting), we can use a pre-trained network to score image-image similarity. Such pre-trained

rewards are typically available across datasets, and monotone with respect to the intuitive notion of *agreement* with $\mathbf{x}_0$. In this sense, they serve as practically justified surrogates for comparing $p_{0|S}$ without requiring an explicit conditional density model.

Our key modeling choice is to use $r(\cdot, \mathbf{s})$ to implicitly characterize $p_{0|S}(\cdot \mid \mathbf{s})$ by tilting the unconditional prior $p_0$ toward higher-reward regions. Our approach is inspired by the alignment framework used in LLMs (Ouyang et al., 2022; Rafailov et al., 2023), where the goal is to generate a sample $\mathbf{x}$ that maximizes some reward $r(\mathbf{x})$, while ensuring that the sampling distribution does not deviate too much from the pre-trained distribution $p_0$. This is typically formalized as a KL-regularized reward maximization problem, $\max_{p \in \mathcal{P}} \left( \mathbb{E}_{\mathbf{x} \sim p}[r(\mathbf{x})] - \tau D_{\mathrm{KL}}(p \| p_0) \right)$, where $\tau > 0$ offers the trade-off between the deviation from the prior and reward maximization. This optimization problem admits a closed-form solution, $p^*(\mathbf{x}) \propto p_0(\mathbf{x}) \exp(r(\mathbf{x})/\tau)$ (Rafailov et al., 2023). Based on this intuition, we make the following *modeling assumption*: the conditional distribution $p_{0|S}$ is approximated as,

$$p_{0|S}(\mathbf{x}_0 \mid \mathbf{s}) \propto p_0(\mathbf{x}_0) \exp\left( \tfrac{r(\mathbf{x}_0; \mathbf{s})}{\tau} \right), \tag{2}$$

This assumption: $(i)$ preserves the powerful unconditional diffusion prior $p_0$, $(ii)$ injects modality-agnostic side information via a reward, and $(iii)$ produces a tractable objective that we can combine with the measurement model to target $p_{0|Y,S}$ at inference time using a pre-trained diffusion model. We do not claim optimality of Eq. (2); rather, we show it leads to a practical, training-free algorithm that consistently improves reconstructions over strong baselines while keeping compute comparable.

We now leverage Eq. (2) to compute the conditional posteriors for the reverse diffusion.

**Proposition 1.** *Let $p_{t|t+1,Y,S}$ denote the conditional posterior distribution for the reverse diffusion process. Then using* (2) *we have*

$$p_{t|t+1,Y,S}(\mathbf{x}_t \mid \mathbf{x}_{t+1}, \mathbf{y}, \mathbf{s}) \propto p_{t|t+1,Y}(\mathbf{x}_t \mid \mathbf{x}_{t+1}, \mathbf{y}) \exp(V_t^\tau(\mathbf{x}_t; \mathbf{s}, \mathbf{y})), \tag{3}$$

$$p_{t|Y,S}(\mathbf{x}_t \mid \mathbf{y}, \mathbf{s}) \propto p_{t|Y}(\mathbf{x}_t \mid \mathbf{y}) \exp(V_t^\tau(\mathbf{x}_t; \mathbf{s}, \mathbf{y})), \tag{4}$$

*where* $V_t^\tau(\mathbf{x}_t; \mathbf{s}, \mathbf{y}) \triangleq \log \mathbb{E}_{\mathbf{x}_0 \sim p_{0|t,Y}(\cdot|\mathbf{x}_t, \mathbf{y})}[\exp(r(\mathbf{x}_0; \mathbf{s})/\tau)]$.

The proof is provided in Appendix A.1. Using (4), we can get the conditional score function as,

$$\nabla_{\mathbf{x}_t} \log p_{t|Y,S}(\mathbf{x}_t \mid \mathbf{y}, \mathbf{s}) = \nabla_{\mathbf{x}_t} \log p_t(\mathbf{x}_t) + \nabla_{\mathbf{x}_t} \log p_{Y|t}(\mathbf{y} \mid \mathbf{x}_t) + \nabla_{\mathbf{x}_t} V_t^\tau(\mathbf{x}_t; \mathbf{s}, \mathbf{y}). \tag{5}$$

The computation of $V_t^\tau$ is not straightforward. So, we use a DPS-style approximation as $V_t^\tau(\mathbf{x}_t; \mathbf{s}, \mathbf{y}) = \log \mathbb{E}_{\mathbf{x}_0 \sim p_{0|t,Y}(\cdot|\mathbf{x}_t, \mathbf{y})}[\exp(r(\mathbf{x}_0; \mathbf{s})/\tau)] \approx r(\mathbb{E}_{\mathbf{x}_0 \sim p_{0|t,Y}(\cdot|\mathbf{x}_t, \mathbf{y})}[\mathbf{x}_0]; \mathbf{s})/\tau = r(\hat{\mathbf{x}}_{0|t,Y}(\mathbf{x}_t, \mathbf{y}); \mathbf{s})/\tau$. Using some approximation and the fact that $p_{Y|0}$ is Gaussian, we can get

$$\hat{\mathbf{x}}_{0|t,Y}(\mathbf{x}_t, \mathbf{y}) \approx \hat{\mathbf{x}}_{0|t}(\mathbf{x}_t) - (1 - \alpha_t)(\sqrt{\alpha_t})\eta \nabla_{\mathbf{x}_t} \|\mathbf{y} - \mathbf{A}\hat{\mathbf{x}}_{0|t}(\mathbf{x}_t)\|_2^2, \tag{6}$$

$$V_t^\tau(\mathbf{x}_t; \mathbf{s}, \mathbf{y}) \approx \hat{V}_t^\tau(\mathbf{x}_t; \mathbf{s}, \mathbf{y}) \triangleq r\left( \hat{\mathbf{x}}_{0|t,Y}(\mathbf{x}_t, \mathbf{y}); \mathbf{s} \right)/\tau. \tag{7}$$

In Appendix A.2, we have provided the details of the steps leading to Eq. (6)-Eq. (7).

We characterize the error in approximating the value function, $|V_t^\tau(\mathbf{x}_t; \mathbf{s}, \mathbf{y}) - \hat{V}_t^\tau(\mathbf{x}_t; \mathbf{s}, \mathbf{y})|$, in Proposition 3, which is deferred to Appendix A.2.

We can now get $\nabla_{\mathbf{x}_t} \log p_{t|Y,S}(\mathbf{x}_t \mid \mathbf{y}, \mathbf{s})$ given in Eq. (5) by replacing $\nabla_{\mathbf{x}_t} V_t^\tau(\mathbf{x}_t; \mathbf{s}, \mathbf{y})$ with $\nabla_{\mathbf{x}_t} \hat{V}_t^\tau(\mathbf{x}_t; \mathbf{y}, \mathbf{s})$. However, running a backward diffusion using $\nabla_{\mathbf{x}_t} \hat{V}_t^\tau(\mathbf{x}_t; \mathbf{s}, \mathbf{y})$ is computationally infeasible because it involves computing second-order derivatives through the denoiser network. This issue, however, can be circumvented by making a further approximation, by setting $\eta = 0$ in Eq. (6) to get $\hat{\mathbf{x}}_{0|t,Y}(\mathbf{x}_t, \mathbf{y}) \approx \hat{\mathbf{x}}_{0|t}(\mathbf{x}_t)$, which leads to the approximation $\nabla_{\mathbf{x}_t} V_t^\tau(\mathbf{x}_t; \mathbf{s}, \mathbf{y}) \approx \nabla_{\mathbf{x}_t} r(\hat{\mathbf{x}}_{0|t}(\mathbf{x}_t); \mathbf{s})$. We show that approximation error remains small when $t$ is small in Appendix A.2 even when $\eta = 0$. This approach then reduces to the **reward gradient guidance (RGG)** approach used for the inference-time alignment of diffusion models (Bansal et al., 2024; Kim et al., 2025b; Yu et al., 2023; He et al., 2024), with the critical difference being that the guidance is from both $\mathbf{s}$ and $\mathbf{y}$.

The RGG approach, however, is limited only to differentiable rewards, and even when they are differentiable, calculating a gradient through the denoiser network at each step of the backward diffusion is computationally intensive and can be ill-suited for many end-use edge-device applications. Moreover, the hyperparameter that determines the weight of the reward gradient guidance is highly sensitive and is difficult to tune, leading to limited performance improvements and undesirable artifacts in the reconstructed images. We later illustrate these issues in Appendix B.6. This motivates us to pursue a gradient-free approach for leveraging the side information for inverse problems.

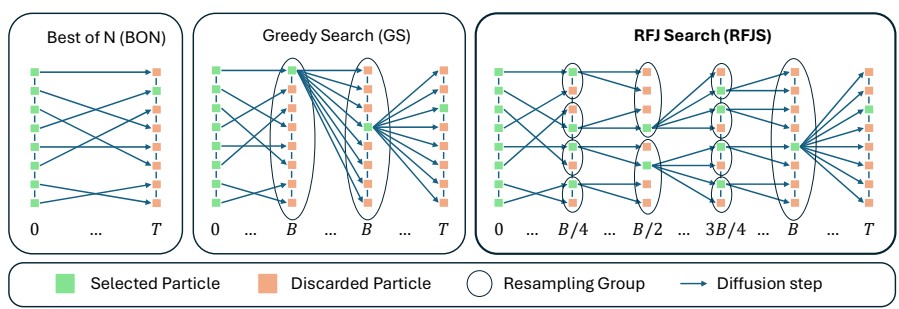

Figure 2: Illustration of the group size resampling strategies of different search algorithms.

## 4.2 INFERENCE-TIME SEARCH ALGORITHMS FOR INVERSE PROBLEMS

Inference-time search methods have recently gained traction as a means of improving the quality of output from LLMs (Snell et al., 2025; Setlur et al., 2025; Liu et al., 2025). The key objective of a search algorithm is to solve a multi-step decision-making problem with balanced exploration and exploitation. While Monte Carlo Tree Search (MCTS) (Kocsis & Szepesvári, 2006) was successful in large-scale reinforcement learning systems like AlphaGo (Silver et al., 2016), they are infeasible for diffusion models: estimating the expected reward of a noisy state $\mathbf{x}_t$ would require repeated rollouts of the reverse process. Training a value function to guide the search rewards is another alternative, but this demands additional training customized to each modality and type of side information. These limitations motivate inference-time methods that are both training-free and computationally tractable.

Particle-based procedures offer one such approach, using the distribution given by Eq. (3), where the value function is replaced by the approximation in Eq. (7). At a given step, suppose we have $N$ samples $\mathbf{x}_{t+1}[1], \ldots, \mathbf{x}_{t+1}[N] \sim p_{t+1|Y,S}$. One way to generate samples from $p_{t|Y,S}$ is to $(i)$ propose candidates $\tilde{\mathbf{x}}_t[i] \sim p_{t|t+1,Y}(\cdot \mid \mathbf{x}_{t+1}[i], \mathbf{y})$, $(ii)$ compute rewards $r[i] = r(\hat{\mathbf{x}}_{0|t,Y}(\tilde{\mathbf{x}}_t[i], \mathbf{y}); \mathbf{s})$ (approximate value) $(iii)$ assign weights $w[i] \propto \exp(r[i]/\tau)$ and resample indices with replacement $I[i] \sim \mathrm{Cat}(w[1:N])$, and $(iv)$ retain $\mathbf{x}_t[I[i]]$ for the next step. In theory, such particle methods converge to the target distribution as $N \to \infty$ and with exact tilting (Wu et al., 2023; Dou & Song, 2024). In practice, however, finite $N$ and approximate tilting has some issues: frequent resampling favors exploitation but risks reward over-optimization, while no resampling preserves data consistency but requires prohibitively many particles to harness and optimize the reward.

To address this trade-off, we modify the step $(iii)$ by introducing **grouped resampling** at each time step $t$, where particles are resampled within groups of size $g_t$. Formally, let the index set of the $i$-th group be $\mathcal{G}_i = \{(i-1)g_t + 1, \ldots, ig_t\}$ for $i = 1, \ldots, N/g_t$. For each group, we draw $g_t$ indices with replacement according to the weights within that groups, i.e., $I'[j] \sim \mathrm{Cat}(w[\mathcal{G}_i]) \in [1:g_t]$. The absolute indices are then obtained by shifting $I'$ as $I[(i-1)g_t + j : ig_t + j] = (i-1)g_t + I'[j]$ for $j = 1, \ldots, g_t$. Based on the choice of $g_t$, we introduce two specific search strategies.

**Greedy Search (GS):** Here, we use a fixed resampling period $B$ and select $g_t = N$ whenever $t \bmod B = 0$, and $g_t = 1$ otherwise. Greedy Search reduces to the **Best-of-N** (BON) strategy when $B \geq T$, since in that case $g_t = 1$ for all $t$. Smaller values of $B$ emphasize short-term reward exploitation, while larger values promote long-term consistency and exploration. An illustration of Greedy Search, with resampling interval $B$, is provided in Figure 2, where the particles evolve independently between resampling events and only interact at steps that are multiples of $B$.

**Recursive Fork-Join Search (RFJS):** Greedy search considers the largest resampling group size $(N)$ at fixed time periods of $B$, and greedily selects one particle from this group, which leads to an exploitation-style approach in search. Selecting the smallest group size $(g_t = 1)$ leads to a pure exploration-style search of BON. Ideally, one should combine the benefits of resampling with multiple group sizes at multiple time steps to get a balanced exploration and exploitation.

To this end, we propose a *recursive grouping and sampling approach*, which we call recursive fork-join search (RFJS), in which the resampling group sizes vary systematically over time. At every $B$ steps, all $N$ particles are resampled together; at every $B/2$ steps, the particles are partitioned into groups of size $N/2$ that are resampled independently; at every $B/4$ steps, groups of size $N/4$ are resampled; and so on. This hierarchical schedule is illustrated in Figure 2. As a concrete example,

Figure 3: **Image as side information:** Qualitative illustration of the performance of our RFJS algorithm compared to the DPS baseline on linear and nonlinear inverse problems. RFJS is able to capture many details that are missed by the DPS baseline to achieve a superior reconstruction quality.

| Algorithm | Box Inpainting | | | | Super Resolution ($\times 4$) | | | | Non-linear Deblur | | | |
|---|---|---|---|---|---|---|---|---|---|---|---|---|
| | FS ($\downarrow$) | PSNR ($\uparrow$) | LPIPS ($\downarrow$) | SSIM ($\uparrow$) | FS ($\downarrow$) | PSNR ($\uparrow$) | LPIPS ($\downarrow$) | SSIM ($\uparrow$) | FS ($\downarrow$) | PSNR ($\uparrow$) | LPIPS ($\downarrow$) | SSIM ($\uparrow$) |
| RFJS (ours) | **0.308** | **28.29** | **0.136** | **0.855** | **0.380** | **25.26** | **0.225** | 0.695 | **0.394** | 23.89 | **0.229** | 0.668 |
| GS (ours) | 0.349 | 28.22 | 0.137 | 0.855 | 0.460 | 25.24 | 0.225 | **0.696** | 0.467 | 23.92 | 0.232 | **0.669** |
| RGG | 0.475 | 27.96 | 0.138 | 0.851 | 0.573 | 25.13 | 0.228 | 0.690 | 0.654 | 23.89 | 0.231 | 0.665 |
| BON | 0.584 | 28.20 | 0.137 | 0.854 | 0.915 | 25.14 | 0.229 | 0.694 | 0.881 | 23.89 | 0.233 | 0.667 |
| DPS | 0.739 | 27.93 | 0.139 | 0.852 | 1.042 | 25.13 | 0.229 | 0.693 | 1.008 | 23.87 | 0.232 | 0.666 |
| | Motion Deblur | | | | Gaussian Deblur | | | | Blind Deblur | | | |
| RFJS (ours) | **0.326** | **26.64** | **0.193** | **0.736** | **0.330** | **26.20** | **0.196** | **0.712** | **0.341** | 25.04 | **0.209** | 0.707 |
| GS (ours) | 0.392 | 26.58 | 0.193 | 0.735 | 0.385 | 26.16 | 0.198 | 0.711 | 0.417 | 25.04 | 0.211 | 0.706 |
| RGG | 0.497 | 26.55 | 0.193 | 0.733 | 0.495 | 26.15 | 0.200 | 0.709 | 0.473 | 24.97 | 0.211 | 0.701 |
| BON | 0.671 | 26.57 | 0.194 | 0.735 | 0.667 | 26.18 | 0.201 | 0.711 | 0.642 | **25.15** | 0.210 | **0.708** |
| DPS | 0.815 | 26.54 | 0.194 | 0.734 | 0.807 | 26.15 | 0.200 | 0.711 | 0.779 | 24.98 | 0.213 | 0.704 |

Table 1: **Image as side information:** Quantitative comparison of our GS and RFJS algorithms with baseline algorithms. For each evaluation metric, the best result is shown in **bold**, and the second best is underlined. RFJS and GS achieve superior performance consistently across all tasks and metrics.

consider $N = 8$. In this case, groups of size at least $N/4 = 2$ are resampled every $B/4$ steps, groups of size $N/2 = 4$ are resampled every $B/2$ steps, and all $N = 8$ particles are resampled every $B$ steps. When multiple group sizes are scheduled to be resampled at the same time step $t$, the larger group size always takes precedence. For example, although $t = B/2$ is also a multiple of $B/4$, the scheme prioritizes the larger group size. Thus, rather than resampling groups of size 2, we resample groups of size 4 at $t = B/2$. Similarly, at $t = B$, the entire set of $N = 8$ particles is resampled jointly. More generally, the group size at time step $t$ is given by $g_t = N/2^{j^*}$, where $j^* = \min\{ i \geq 0 : t \bmod (B/2^i) = 0 \}$. The localized resampling (fork) at intermediate group sizes encourages balanced exploration, while the recursive return to larger group sizes (join) encourages exploitation. Naively reducing $B$ in GS does not balance this trade-off well and may lead to an undesirable compromise between exploration and exploitation.

We have summarized this inference-time search framework in Algorithm 1 in the Appendix A.3. Our framework is modular: the resampling rule, whether BON, GS, or RFJS, can be chosen depending on budget and application. Since this requires no retraining and works with arbitrary reward functions, it can be incorporated into any diffusion-based inverse problem solvers with minimal modification.

## 5 EXPERIMENTS

### 5.1 EXPERIMENTAL SETUP

We evaluate our inference-time search framework for solving inverse problems with side information by instantiating two specific search algorithms we proposed: **Greedy Search (GS)** and **Recursive Fork Join Search (RFJS)**, both of which are described in the previous section. We consider two types of side information: $(i)$ **image as side information**, where a reference image of the same entity (here, the same person under different poses/lighting) is used as side information, and $(ii)$ **text as side information**, where a text description of the target image is used as a side information.

Figure 4: **Text as side information :** Qualitative illustration of the performance of our RFJS algorithm compared to the DPS baseline. For example, the side information provided for the super resolution task is *'golden retriever sitting on a snowy frozen lake, facing forward'*. RFJS is able to capture many details that are missed by the DPS baseline to achieve a superior reconstruction quality.

| | Box Inpainting | | | | Super Resolution ($\times 32$) | | | | Non-linear Deblur | | | |
|---|---|---|---|---|---|---|---|---|---|---|---|---|
| Algorithm | CS ($\uparrow$) | PSNR ($\uparrow$) | SSIM ($\uparrow$) | LPIPS ($\downarrow$) | CS ($\uparrow$) | PSNR ($\uparrow$) | SSIM ($\uparrow$) | LPIPS ($\downarrow$) | CS ($\uparrow$) | PSNR ($\uparrow$) | SSIM ($\uparrow$) | LPIPS ($\downarrow$) |
| RFJS (ours) | **0.901** | **20.75** | **0.678** | **0.294** | **0.801** | 17.13 | **0.352** | 0.4926 | 0.863 | **20.58** | **0.473** | **0.405** |
| GS (ours) | 0.894 | 19.76 | 0.676 | 0.305 | 0.791 | 17.20 | 0.351 | 0.5094 | **0.865** | 20.32 | 0.456 | 0.405 |
| BON | 0.882 | 19.99 | 0.672 | 0.308 | 0.788 | **17.21** | 0.350 | 0.5003 | 0.855 | 20.52 | 0.464 | 0.406 |
| DPS | 0.871 | 19.86 | 0.672 | 0.312 | 0.731 | 16.90 | 0.330 | 0.5220 | 0.839 | 20.55 | 0.469 | 0.409 |

| | Motion Deblur | | | | Gaussian Deblur | | | | Blind Deblur | | | |
|---|---|---|---|---|---|---|---|---|---|---|---|---|
| RFJS (ours) | **0.858** | 18.61 | 0.402 | **0.424** | **0.843** | **18.10** | 0.358 | 0.457 | **0.851** | 18.84 | 0.412 | **0.433** |
| GS (ours) | 0.835 | 17.83 | 0.369 | 0.453 | 0.835 | 17.96 | 0.356 | 0.457 | 0.835 | **18.93** | **0.414** | 0.438 |
| BON | 0.848 | **19.24** | **0.415** | 0.427 | 0.831 | 17.99 | **0.365** | **0.452** | 0.831 | 18.78 | 0.410 | 0.443 |
| DPS | 0.794 | 18.16 | 0.384 | 0.458 | 0.778 | 16.79 | 0.329 | 0.487 | 0.793 | 18.82 | 0.409 | 0.459 |

Table 2: **Text as side information:** Quantitative comparison of our GS and RFJS algorithms with baseline algorithms. RFJS and GS achieve better performance across all tasks and metrics.

We demonstrate the plug-and-play nature of our algorithms by considering four different baseline inverse problem solvers: $(i)$ **DPS** (Chung et al., 2023b), $(ii)$ **BlindDPS** (Chung et al., 2023a), $(iii)$ **MPGD** (He et al., 2024), and $(iv)$ **DAPS** (Zhang et al., 2024). Due to page limitation, the evaluation results using DAPS and MPGD are deferred to Appendix B.

**Inverse problems:** We evaluate our algorithms on six inverse problems, covering both linear and nonlinear problems. The linear problems are: $(i)$ box inpainting, $(ii)$ super resolution, $(iii)$ motion deblurring, and $(iv)$ Gaussian deblurring. The nonlinear problems are: $(v)$ nonlinear deblurring, and $(vi)$ blind deblurring. A detailed description of these inverse problems is given in Appendix C.

**Baselines:** We compare the performance of GS and RFJS against the following baselines: $(i)$ *Baseline solvers (DPS, BlindDPS, MPGD, DAPS)*, $(ii)$ *Best-of-N (BoN)*, which generates $N$ independent samples and selects the one with the best reward at the end, $(iii)$ *Reward Gradient Guidance (RGG)*, which solves the inverse problem by running the backward diffusion according to Eq. (5), but with the approximation $\nabla_{\mathbf{x}_t} V_t^\tau(\mathbf{x}_t; \mathbf{s}, \mathbf{y}) \approx \nabla_{\mathbf{x}_t} r(\hat{\mathbf{x}}_{0|t}(\mathbf{x}_t); \mathbf{s})$. Unless otherwise noted, hyperparameters, including guidance scale, number of diffusion steps, and task-specific settings, match the original baseline implementations. The specific values of hyperparamaters are listed in Appendix C. All the experiments are run on NVIDIA A100 GPUs on an internal compute cluster.

## 5.2 MAIN RESULTS

**Image as side information:** The goal is to reconstruct a human face from a noisy observation when another image of the same identity is available (Fig. 3). Using Celeb-HQ (Na et al., 2022) as an out-of-distribution set and a diffusion model pretrained on FFHQ (Chung et al., 2023b), we sample two random images per identity for target and side information. We compute the reward as follows: first, detect the face using MTCNN (Zhang et al., 2016) and then extract identity features with AdaFace (Kim et al., 2022). Then, we measure the reward as the negative of the FaceSimilarity (FS) loss, computed as the distance between the identity embeddings of the reconstructed and side-information faces, extracted by pretrained AdaFace network. We evaluate on 64 pairs, using $N = 8$ particles and $B = 16$, with a gradient scale $0.5$ for RGG. We evaluate with standard metrics, PSNR, SSIM, and LPIPS, but these often fail to measure the identity similarity. Thus, we use FaceSimilarity (FS),

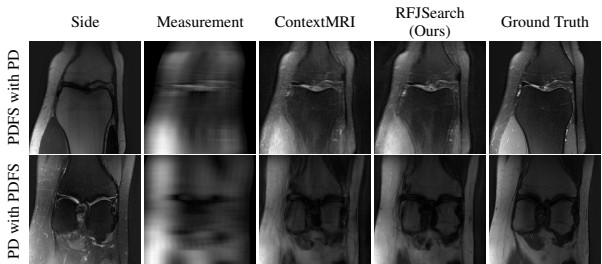

Figure 5: **Contrast Image as Side Information:** Qualitative MRI reconstruction with RFJS vs. ContextMRI. The shapes and line edges are well preserved in our reconstruction.

Figure 6: Quantitative MRI reconstruction results (fastMRI knee, AF=16, ACS=2%).

| Algorithm | PSNR (↑) | SSIM (↑) | LPIPS (↓) | NMI (↑) |
|---|---|---|---|---|
| PDFS with PD | | | | |
| RFJS | **25.85** | **0.801** | **0.375** | **0.457** |
| GS | 25.33 | 0.797 | 0.375 | 0.455 |
| BON | 25.47 | 0.797 | 0.376 | 0.454 |
| ContextMRI | 25.39 | 0.795 | 0.383 | 0.451 |
| PD with PDFS | | | | |
| RFJS | **27.85** | **0.920** | **0.358** | **0.579** |
| GS | 27.80 | **0.920** | 0.360 | 0.579 |
| BON | 27.80 | 0.918 | 0.366 | 0.570 |
| ContextMRI | 27.46 | 0.915 | 0.375 | 0.563 |

comparing the reconstruction to the ground truth for a more reliable measure of identity preservation. Table 1 shows that both proposed inference-time search methods, GS and RFJS, outperform baselines, with RFJS achieving the best overall scores indicating a stronger balance between exploration and exploitation. Qualitative results given in Fig 3 show sharper facial details and preserved identity traits, whereas Fig 7 and a detailed discussion in Appendix B.1 indicates the importance of the FS metric.

**Text as side information:** The goal is to reconstruct an image from its noisy observation, with a text description of the image available as side information. We use a pre-trained diffusion model trained on the ImageNet data (Dhariwal & Nichol, 2021). We use 25 images from the ImageNet validation set to evaluate the algorithms and generated a short one-sentence textual description for each image using ChatGPT. We use ImageReward (Xu et al., 2023), a pre-trained network that measures text-to-image similarity, as the reward function. We consider some inverse problem tasks that are significantly challenging, including $\times 32$ super resolution, and strong blur with larger kernels. Experiments use $N$=4 and $B$=100, and we report the standard metrics and CLIPScore (Radford et al., 2021). CLIPScore measures the cosine similarity between CLIP image embeddings of the ground truth and reconstruction, providing a semantically informed metric that reflects both visual and textual alignment. It can be seen in Fig. 4 that the qualitative reconstructions closely match textual descriptions. The quantitative metrics are in Table 2 where both GS and RFJS outperform competing baselines, with RFJS achieving the highest CLIPScore.

**MRI with multi-contrast side information:** Finally, we test on fastMRI knee dataset (Zbontar et al., 2018) with the ContextMRI model (Chung et al., 2025). We pair PD and PDFS contrasts, reconstructing one from the other under highly accelerated $16\times$ undersampling with 2% ACS. We use normalized mutual information (NMI) as reward, which is robust to contrast changes. Table 6 shows our methods consistently outperform the baseline in all the metrics of interest. Figure 5 highlights sharper edges and more faithful structure.

**Additional Experiments:** To demonstrate the generality of our framework, we extended our search algorithm beyond DPS to other samplers, including DAPS and MPGD. Qualitative and quantitative results for these experiments are provided in Appendix B, along with additional DPS results for both types of side information. We also conducted several ablations to analyze the role of side information and the scalability of our approach. Appendix B.6 studies the sensitivity of the gradient-guided methods. Appendix B.7 examines the effect of the number of particles: increasing $N$ improves exploration and reward, while runtime grows sublinearly due to parallelization (Appendix B.8). To build intuition, Appendix B.9 provides 2D toy examples illustrating the benefits of side information and the impact of $B$; Fig. 25 further shows that RFJS is more robust than GS when the reward is non-linear and non-convex. Hyperparameter details are summarized in Appendix C.

## 6 CONCLUSION

We proposed a lightweight, modular inference-time search algorithm that integrates side information into diffusion-based image reconstruction, in a principled way. By adaptively guiding the generative process, our method delivers substantial quality gains, especially in ill-posed settings, while requiring only minimal changes to existing pipelines. Extensive experiments across standard reconstruction tasks show consistent improvements in both visual fidelity and quantitative metrics, and our approach surpasses gradient-based alternatives. These results highlight the power of leveraging side information at inference time to make diffusion-based solvers more reliable and accurate.

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
