## APPENDICES

## A PROOFS

### A.1 PROOF OF PROPOSITION 1

To begin, recall that $p_{0|S}(\mathbf{x}_0|\mathbf{s}) = \frac{1}{Z}p_0(\mathbf{x}_0)e^{\frac{r(\mathbf{x}_0;\mathbf{s})}{\tau}}$. Using Bayes' rule, we can rewrite this expression as

$$\frac{1}{Z}p_0(\mathbf{x}_0)\exp\left(\frac{r(\mathbf{x}_0;\mathbf{s})}{\tau}\right) = \frac{p_{0,S}(\mathbf{x}_0,\mathbf{s})}{p_S(\mathbf{s})} = \frac{p_{S|0}(\mathbf{s}\mid\mathbf{x}_0)p_0(\mathbf{x}_0)}{p_S(\mathbf{s})}$$

and we gather that, for $\mathbf{s}$ fixed, $p_{S|0}(\mathbf{s}|\mathbf{x}_0) \propto e^{\frac{r(\mathbf{x}_0;\mathbf{s})}{\tau}}$ where the proportionality is up to constants on independent of $\mathbf{x}_0$. Starting from the LHS of (3), we first apply Bayes' rule, reverse the conditioning,

and introduce a marginalized $\mathbf{x}_0$ to leverage conditional independence. This sequence leads to

$$p_{t|t+1,Y,S}(\mathbf{x}_t \mid \mathbf{x}_{t+1}, \mathbf{y}, \mathbf{s}) = \frac{p_{t,S|t+1,Y}(\mathbf{x}_t, \mathbf{s} \mid \mathbf{x}_{t+1}, \mathbf{y})}{p_{S|t+1,Y}(\mathbf{s} \mid \mathbf{x}_{t+1}, \mathbf{y})}$$

$$= \frac{p_{t|t+1,Y}(\mathbf{x}_t \mid \mathbf{x}_{t+1}, \mathbf{y})p_{S|t,t+1,Y}(\mathbf{s} \mid \mathbf{x}_t, \mathbf{x}_{t+1}, \mathbf{y})}{p_{S|t+1,Y}(\mathbf{s} \mid \mathbf{x}_{t+1}, \mathbf{y})}$$

$$\propto p_{t|t+1,Y}(\mathbf{x}_t \mid \mathbf{x}_{t+1}, \mathbf{y}) \int_{\mathbf{x}_0} p_{S|t,t+1,Y}(\mathbf{s} \mid \mathbf{x}_0, \mathbf{x}_t, \mathbf{x}_{t+1}, \mathbf{y})p_{0|t,t+1,Y}(\mathbf{x}_0 \mid \mathbf{x}_t, \mathbf{x}_{t+1}, \mathbf{y})d\mathbf{x}_0$$

$$= p_{t|t+1,Y}(\mathbf{x}_t \mid \mathbf{x}_{t+1}, \mathbf{y}) \int_{\mathbf{x}_0} p_{S|0,t,t+1,Y}(\mathbf{s} \mid \mathbf{x}_0, \mathbf{x}_t, \mathbf{x}_{t+1}, \mathbf{y})p_{0|t,Y}(\mathbf{x}_0 \mid \mathbf{x}_t, \mathbf{y})d\mathbf{x}_0$$

$$\propto p_{t|t+1,Y}(\mathbf{x}_t \mid \mathbf{x}_{t+1}, \mathbf{y}) \int_{\mathbf{x}_0} p_{S|0}(\mathbf{s} \mid \mathbf{x}_0)p_{0|t,Y}(\mathbf{x}_0 \mid \mathbf{x}_t, \mathbf{y})d\mathbf{x}_0$$

$$= p_{t|t+1,Y}(\mathbf{x}_t \mid \mathbf{x}_{t+1}, \mathbf{y})\mathbb{E}_{\mathbf{x}_0 \sim p_{0|t,Y}(\cdot|\mathbf{x}_t, \mathbf{y})}[\exp(r(\mathbf{x}_0; \mathbf{s})/\tau)]$$

$$\propto p_{t|t+1,Y}(\mathbf{x}_t \mid \mathbf{x}_{t+1}, \mathbf{y}) \exp(V_t^{\tau}(\mathbf{x}_t; \mathbf{s}, \mathbf{y})).$$

The penultimate step follows from the discussion at the onset of the proof. The last step captures the definition for $V_t^{\tau}(\mathbf{x}_t; \mathbf{s}, \mathbf{y})$ found in Proposition 1. The proof of (4) is similar without conditioning on $t + 1$.

*Proof of Value-titled KL.* Given a distribution $p_0$ over $\mathbb{R}^d$, a reward function $r : \mathbb{R}^d \to \mathbb{R}$, and $\tau > 0$, we are interested in sampling from the distribution $p^*$ given by

$$p^* = \arg\max_p \mathbb{E}_{\mathbf{x} \sim p}[r(\mathbf{x})] - \tau D_{\mathrm{KL}}(p\|p_0)$$

$$= \arg\max_p \mathbb{E}_{\mathbf{x} \sim p}\left[r(\mathbf{x}) - \tau \log \frac{p(\mathbf{x})}{p_0(\mathbf{x})}\right]$$

$$= \arg\min_p \mathbb{E}_{\mathbf{x} \sim p}\left[\log \frac{p(\mathbf{x})}{p_0(x)} - \frac{r(\mathbf{x})}{\tau}\right]$$

$$= \arg\min_p \mathbb{E}_{x \sim p}\left[\log \frac{p(\mathbf{x})}{p_0(\mathbf{x})e^{r(\mathbf{x})/\tau}}\right] \triangleq \arg\min_p \mathcal{L}(p).$$

Let $q(\mathbf{x}) \triangleq \frac{1}{Z}p_0(\mathbf{x})e^{r(\mathbf{x})/\tau}$, where $Z$ is chosen such that $\int q(\mathbf{x})d\mathbf{x} = 1$. Then

$$\mathcal{L}(p) = \mathbb{E}_{\mathbf{x} \sim p}\left[\log \frac{p(\mathbf{x})}{p_0(\mathbf{x})e^{r(\mathbf{x})/\tau}}\right] = \mathbb{E}_{\mathbf{x} \sim p}\left[\log \frac{p(\mathbf{x})}{Zq(\mathbf{x})}\right] = D_{\mathrm{KL}}(p\|q) - \log Z.$$

By the non-negativity of KL-divergence, $\mathcal{L}(p) \geq \mathcal{L}(q)$ for any distribution $p$, and so $p^* = q$, or $p(\mathbf{x}) \propto p_0(\mathbf{x})e^{r(\mathbf{x})/\tau}$.

### A.2 VALUE APPROXIMATION BOUND

In the following, we provide the steps that lead to Eq. 7, and subsequently bound the approximation error. We begin with the following lemma.

**Lemma 2.** *The conditional mean of $X_0$ given $X_t = \mathbf{x}_t$ and $Y = \mathbf{y}$ is given by*

$$\hat{\mathbf{x}}_{0|t,Y}(\mathbf{x}_t, \mathbf{y}) = \hat{\mathbf{x}}_{0|t}(\mathbf{x}_t) + \left(\frac{1 - \alpha_t}{\sqrt{\alpha_t}}\right)\nabla_{\mathbf{x}_t} \log p_{Y|t}(\mathbf{y} \mid \mathbf{x}_t). \tag{8}$$

*Proof.* For any distribution over $X_0$ since $p_{t|0}(\mathbf{x}_t \mid \mathbf{x}_0) = \mathcal{N}(\mathbf{x}_t \mid \sqrt{\alpha_t}\mathbf{x}_0, (1 - \alpha_t)\mathbf{I})$, we can use the Tweedie's formula (Efron, 2011) to $p_{0|Y}(\mathbf{x}_0 \mid \mathbf{y})$ and $p_0(\mathbf{x}_0)$ to get

$$\sqrt{\alpha_t}\hat{\mathbf{x}}_{0|t,Y}(\mathbf{x}_t, \mathbf{y}) = \mathbf{x}_t + (1 - \alpha_t)\nabla_{\mathbf{x}_t} \log p_{t|Y}(\mathbf{x}_t \mid \mathbf{y}) \tag{9}$$

$$\sqrt{\alpha_t}\hat{\mathbf{x}}_{0|t}(\mathbf{x}_t) = \mathbf{x}_t + (1 - \alpha_t)\nabla_{\mathbf{x}_t} \log p_t(\mathbf{x}_t). \tag{10}$$

Since by Bayes theorem, $\nabla_{\mathbf{x}_t} \log p_{t|Y}(\mathbf{x}_t \mid \mathbf{y}) = \nabla_{\mathbf{x}_t} \log p_t(\mathbf{x}_t) + \log p_{Y|t}(\mathbf{y} \mid \mathbf{x}_t)$, the results follows by simple algebra.

Since $p_{Y|0}$ is Gaussian, using the DPS approximation on the second term in Eq. (8), we get

$$\hat{\mathbf{x}}_{0|t,Y}(\mathbf{x}_t, \mathbf{y}) \approx \hat{\mathbf{x}}_{0|t}(\mathbf{x}_t) - \left(\frac{1-\alpha_t}{\sqrt{\alpha_t}}\right) \frac{1}{2\sigma_y^2} \nabla_{\mathbf{x}_t} \|\mathbf{y} - \mathbf{A}(\hat{\mathbf{x}}_{0|t}(\mathbf{x}_t))\|_2^2. \tag{11}$$

Replacing $1/2\sigma_y^2$ by $\eta$ to control the approximation error gives Eq. (6). In the following, we denote this approximation as

$$\hat{\mathbf{x}}_{0|t,Y}(\mathbf{x}_t, \mathbf{y}) \approx \hat{\mathbf{x}}_{0|t}(\mathbf{x}_t) - \left(\frac{1-\alpha_t}{\sqrt{\alpha_t}}\right) \eta \nabla_{\mathbf{x}_t} \|\mathbf{y} - \mathbf{A}(\hat{\mathbf{x}}_{0|t}(\mathbf{x}_t))\|_2^2 \triangleq \tilde{\mathbf{x}}_{0|t}^\eta(\mathbf{x}_t, \mathbf{y}). \tag{12}$$

**Proposition 3.** *Assume that $r$ is a Lipschitz function that takes values in $[0,1]$. For any $\mathbf{x}_t, \mathbf{y}, \mathbf{s}$, the error in the value approximation $\hat{V}_t^\tau(\mathbf{x}_t; \mathbf{s}, \mathbf{y}) = r(\tilde{\mathbf{x}}_{0|t}^\eta(\mathbf{x}_t, \mathbf{y}); \mathbf{s})/\tau$ with the true value $V_t^\tau(\mathbf{x}_t; \mathbf{s}, \mathbf{y})$ is bounded as*

$$|V_t^\tau(\mathbf{x}_t; \mathbf{s}, \mathbf{y}) - \hat{V}_t^\tau(\mathbf{x}_t; \mathbf{s}, \mathbf{y})| \leq c_\tau c_1(t) + \frac{L_r}{\tau}\left(\sqrt{c_2(t)} + \sqrt{c_3(t)c_4^\eta(t)}\right), \tag{13}$$

*where $c_1(t) = \text{Var}\left(r(X_0; \mathbf{s}) \mid \mathbf{x}_t, \mathbf{y}\right)$, $c_2(t) = \text{Var}(X_0 \mid \mathbf{x}_t, \mathbf{y})$, $c_3(t) = \text{Var}(X_0 \mid \mathbf{x}_t)$,*

$$c_4^\eta(t) = 1 + \text{CV}^2(t) + \eta^2 \|\mathbf{A}^T(\mathbf{y} - \mathbf{A}\hat{\mathbf{x}}_{0|t}(\mathbf{x}_t))\|_{\Sigma_{0|t}(\mathbf{x}_t)}^2 \tag{14}$$

$$- 2\eta\langle\mathbf{A}(\hat{\mathbf{x}}_{0|t,Y}(\mathbf{x}_t, \mathbf{y}) - \hat{\mathbf{x}}_{0|t}(\mathbf{x}_t)), \mathbf{y} - \mathbf{A}\hat{\mathbf{x}}_{0|t}(\mathbf{x}_t)\rangle, \tag{15}$$

*where $\text{CV}(t) \triangleq \frac{\sqrt{\text{Var}(p_{Y|0}(\mathbf{y}|X_0)|\mathbf{x}_t)}}{\mathbb{E}[p_{Y|0}(\mathbf{y}|X_0)|\mathbf{x}_t]}$ is the coefficient of variation of the likelihood function $p_{Y|0}(\mathbf{y} \mid X_0)$ given $\mathbf{x}_t$ and $c_\tau = e^{1/\tau} - 1 - 1/\tau$ is a positive constant.*

**Remark 4.** *In Proposition 3, the term $c_1(t)$ denotes the conditional variance of the reward given $\mathbf{x}_t, \mathbf{y}$, $c_2(t)$ denotes the conditional variance of $X_0$ given $\mathbf{x}_t, \mathbf{y}$, and $c_3(t)$ denotes the conditional variance of $X_0$ given only $\mathbf{x}_t$. Since the variance of the reverse distribution $p_{0|t}(\cdot \mid \mathbf{x}_t)$ decreases as $t$ becomes smaller, we have that all the terms $c_1(t), c_2(t), c_3(t)$ are small when $t$ is small. Therefore, the approximation error is small when $t$ is small.*

*Proof.* Since $r$ is a bounded random variable, assuming finite variance, we can use Bennett's inequality for the log moment-generating function

$$V_t^\tau(\mathbf{x}_t; \mathbf{s}, \mathbf{y}) = \log \mathbb{E}[\exp(r(X_0; \mathbf{s})/\tau)] \leq \frac{1}{\tau}\mathbb{E}[r(X_0; \mathbf{s})] + c_\tau c_1(t). \tag{16}$$

Then, we have

$$\left|\log \mathbb{E}[\exp(r(X_0; \mathbf{s})/\tau)] - \frac{1}{\tau}r(\tilde{\mathbf{x}}_{0|t,Y}^\eta(\mathbf{x}_t, \mathbf{y}))\right| \leq \frac{1}{\tau}|\mathbb{E}[r(X_0; \mathbf{s})] - r(\tilde{\mathbf{x}}_{0|t,Y}^\eta(\mathbf{x}_t, \mathbf{y}))| + c_\tau c_1(t). \tag{17}$$

Now, let us simplify the first term,

$$|\mathbb{E}[r(X_0; \mathbf{s})] - r(\tilde{\mathbf{x}}_{0|t,Y}^\eta(\mathbf{x}_t, \mathbf{y}); \mathbf{s})| \leq \mathbb{E}[|r(X_0; \mathbf{s}) - r(\tilde{\mathbf{x}}_{0|t,Y}^\eta(\mathbf{x}_t, \mathbf{y}); \mathbf{s})|] \tag{18}$$

$$\leq L_r \mathbb{E}[\|X_0 - \tilde{\mathbf{x}}_{0|t,Y}^\eta(\mathbf{x}_t, \mathbf{y})\|_2] \tag{19}$$

$$\leq L_r(\mathbb{E}[\|X_0 - \hat{\mathbf{x}}_{0|t,Y}(\mathbf{x}_t, \mathbf{y})\|_2] + \|\hat{\mathbf{x}}_{0|t,Y}(\mathbf{x}_t, \mathbf{y}) - \tilde{\mathbf{x}}_{0|t,Y}^\eta(\mathbf{x}_t, \mathbf{y})\|_2). \tag{20}$$

The first term can be bounded by $\sqrt{c_2(t)}$ using Cauchy-Schwarz inequality in $L^2$-probability space. For the second term, first we simplify

$$\tilde{\mathbf{x}}_{0|t,Y}^\eta(\mathbf{x}_t, \mathbf{y}) = \hat{\mathbf{x}}_{0|t}(\mathbf{x}_t) - \left(\frac{1-\alpha_t}{\sqrt{\alpha_t}}\right) \eta \nabla_{\mathbf{x}_t} \|\mathbf{y} - \mathbf{A}\hat{\mathbf{x}}_{0|t}(\mathbf{x}_t)\|_2^2,$$

and then $\nabla_{\mathbf{x}_t} \|\mathbf{y} - \mathbf{A}\hat{\mathbf{x}}_{0|t}(\mathbf{x}_t)\|_2^2 = -(\nabla_{\mathbf{x}_t}\hat{\mathbf{x}}_{0|t}(\mathbf{x}_t))\mathbf{A}^T(\mathbf{y} - \mathbf{A}\hat{\mathbf{x}}_{0|t}(\mathbf{x}_t))$. Now,

$$\nabla_{\mathbf{x}_t}\hat{\mathbf{x}}_{0|t}(\mathbf{x}_t) = \nabla_{\mathbf{x}_t}\int \mathbf{x}_0^T p_{0|t}(\mathbf{x}_0 \mid \mathbf{x}_t)\mathrm{d}\mathbf{x}_0 = \int \nabla_{\mathbf{x}_t} p_{0|t}(\mathbf{x}_0 \mid \mathbf{x}_t)\mathbf{x}_0^T \mathrm{d}\mathbf{x}_0 \tag{21}$$

$$= \int \nabla_{\mathbf{x}_t} \log p_{0|t}(\mathbf{x}_0 \mid \mathbf{x}_t)\mathbf{x}_0^T p_{0|t}(\mathbf{x}_0 \mid \mathbf{x}_t)\mathrm{d}\mathbf{x}_0. \tag{22}$$

Now, we shall compute

$$\nabla_{\mathbf{x}_t} \log p_{0|t}(\mathbf{x}_0 \mid \mathbf{x}_t) = \nabla_{\mathbf{x}_t} \log p_{t|0}(\mathbf{x}_t \mid \mathbf{x}_0) - \nabla_{\mathbf{x}_t} \log p_t(\mathbf{x}_t). \tag{23}$$

But since $\sqrt{\alpha_t}\hat{\mathbf{x}}_{0|t}(\mathbf{x}_t) = \mathbf{x}_t + (1-\alpha_t)\nabla_{\mathbf{x}_t} \log p_t(\mathbf{x}_t)$, and $\nabla_{\mathbf{x}_t} \log p_{t|0}(\mathbf{x}_t \mid \mathbf{x}_0) = \frac{1}{1-\alpha_t}(\sqrt{\alpha_t}\mathbf{x}_0 - \mathbf{x}_t)$, which gives

$$\left(\frac{1-\alpha_t}{\sqrt{\alpha_t}}\right)\nabla_{\mathbf{x}_t} \log p_{0|t}(\mathbf{x}_0 \mid \mathbf{x}_t) = \mathbf{x}_0 - \hat{\mathbf{x}}_{0|t}(\mathbf{x}_t), \tag{24}$$

which gives

$$\left(\frac{1-\alpha_t}{\sqrt{\alpha_t}}\right)\nabla_{\mathbf{x}_t}\hat{\mathbf{x}}_{0|t}(\mathbf{x}_t) = \mathbb{E}_{X_0 \sim p_{0|t}(\mathbf{x}_t)}[(X_0 - \hat{\mathbf{x}}_{0|t}(\mathbf{x}_t))X_0^T] \tag{25}$$

$$= \mathbb{E}_{p_{0|t}(\mathbf{x}_t)}[X_0 X_0^T] - \mathbb{E}_{p_{0|t}(\mathbf{x}_t)}[X_0]\mathbb{E}_{p_{0|t}(\mathbf{x}_t)}[X_0^T] \tag{26}$$

$$= \mathbb{E}_{p_{0|t}(\mathbf{x}_t)}[(X_0 - \hat{\mathbf{x}}_{0|t}(\mathbf{x}_t))(X_0 - \hat{\mathbf{x}}_{0|t}(\mathbf{x}_t))^T] \triangleq \Sigma_{0|t}(\mathbf{x}_t), \tag{27}$$

which is precisely the covariance matrix of $X_0$ given $\mathbf{x}_t$.

Thus, we finally get,

$$\tilde{\mathbf{x}}_{0|t,Y}^{\eta}(\mathbf{x}_t, \mathbf{y}) = \hat{\mathbf{x}}_{0|t}(\mathbf{x}_t) + \eta\Sigma_{0|t}(\mathbf{x}_t)\mathbf{A}^T(\mathbf{y} - \mathbf{A}\hat{\mathbf{x}}_{0|t}(\mathbf{x}_t)) \tag{28}$$

Next, note that since $\mathbb{E}_{p_{0|t}(\mathbf{x}_t)}[p_{Y|0}(\mathbf{y} \mid X_0)] = p_{Y|t}(\mathbf{y} \mid \mathbf{x}_t)$, we can define $f(X_0) = \frac{p_{Y|0}(\mathbf{y}|X_0)}{\mathbb{E}_{p_{0|t}(\mathbf{x}_t)}[p_{Y|0}(\mathbf{y}|X_0)]}$, whose expectation is $\mathbb{E}_{p_{0|t}(\mathbf{x}_t)}[f(X_0)] = 1$. Further, it is easy to see that $\hat{\mathbf{x}}_{0|t,Y}(\mathbf{x}_t, \mathbf{y}) = \mathbb{E}_{p_{0|t}(\mathbf{x}_t)}[X_0 f(X_0)]$. Now, we are ready to bound the final term as follows

$$\|\hat{\mathbf{x}}_{0|t,Y}(\mathbf{x}_t, \mathbf{y}) - \tilde{\mathbf{x}}_{0|t,Y}^{\eta}(\mathbf{x}_t, \mathbf{y})\|_2 \tag{29}$$

$$= \|\mathbb{E}_{p_{0|t}(\mathbf{x}_t)}[X_0 f(X_0)] - \hat{\mathbf{x}}_{0|t}(\mathbf{x}_t) - \eta\Sigma_{0|t}(\mathbf{x}_t)\mathbf{A}^T(\mathbf{y} - \mathbf{A}\hat{\mathbf{x}}_{0|t}(\mathbf{x}_t))\|_2 \tag{30}$$

$$= \|\mathbb{E}_{p_{0|t}(\mathbf{x}_t)}[X_0 f(X_0) - \hat{\mathbf{x}}_{0|t}(\mathbf{x}_t)f(X_0)] \tag{31}$$

$$- \eta\mathbb{E}_{p_{0|t}(\mathbf{x}_t)}[(X_0 - \hat{\mathbf{x}}_{0|t}(\mathbf{x}_t))(X_0 - \hat{\mathbf{x}}_{0|t}(\mathbf{x}_t))^T]\mathbf{A}^T(\mathbf{y} - \mathbf{A}\hat{\mathbf{x}}_{0|t}(\mathbf{x}_t))\|_2 \tag{32}$$

$$= \|\mathbb{E}_{p_{0|t}(\mathbf{x}_t)}[(X_0 - \hat{\mathbf{x}}_{0|t}(\mathbf{x}_t))(f(X_0) - \eta(X_0 - \hat{\mathbf{x}}_{0|t}(\mathbf{x}_t))^T\mathbf{A}^T(\mathbf{y} - \mathbf{A}\hat{\mathbf{x}}_{0|t}(\mathbf{x}_t)))]\|_2 \tag{33}$$

$$\leq \mathbb{E}_{p_{0|t}(\mathbf{x}_t)}[\|(X_0 - \hat{\mathbf{x}}_{0|t}(\mathbf{x}_t))(f(X_0) - \eta(X_0 - \hat{\mathbf{x}}_{0|t}(\mathbf{x}_t))^T\mathbf{A}^T(\mathbf{y} - \mathbf{A}\hat{\mathbf{x}}_{0|t}(\mathbf{x}_t)))\|_2] \tag{34}$$

$$= \mathbb{E}_{p_{0|t}(\mathbf{x}_t)}[\|X_0 - \hat{\mathbf{x}}_{0|t}(\mathbf{x}_t)\|_2 |f(X_0) - \eta(X_0 - \hat{\mathbf{x}}_{0|t}(\mathbf{x}_t))^T\mathbf{A}^T(\mathbf{y} - \mathbf{A}\hat{\mathbf{x}}_{0|t}(\mathbf{x}_t))|] \tag{35}$$

$$\leq \sqrt{c_3(t)}\sqrt{c_4^{\eta}(t)}, \tag{36}$$

where the last step follows by Cauchy-Schwarz inequality in $L^2$ probability space, where

$$c_3(t) \triangleq \mathbb{E}_{p_{0|t}(\mathbf{x}_t)}[\|X_0 - \mathbb{E}_{p_{0|t}(\mathbf{x}_t)[X_0]}\|_2^2] = \text{Var}(X_0 \mid \mathbf{x}_t). \tag{37}$$

and at last, we have

$$c_4^{\eta}(t) \triangleq \mathbb{E}_{p_{0|t}(\mathbf{x}_t)}[(f(X_0) - \eta(X_0 - \hat{\mathbf{x}}_{0|t}(\mathbf{x}_t))^T\mathbf{A}^T(\mathbf{y} - \mathbf{A}\hat{\mathbf{x}}_{0|t}(\mathbf{x}_t)))^2] \tag{38}$$

$$= \mathbb{E}_{p_{0|t}(\mathbf{x}_t)}[f(X_0)^2] + \eta^2\mathbb{E}[((X_0 - \hat{\mathbf{x}}_{0|t}(\mathbf{x}_t))^T\mathbf{A}^T(\mathbf{y} - \mathbf{A}\hat{\mathbf{x}}_{0|t}(\mathbf{x}_t)))^2] \tag{39}$$

$$- 2\eta\mathbb{E}_{p_{0|t}(\mathbf{x}_t)}[(X_0 f(X_0) - f(X_0)\hat{\mathbf{x}}_{0|t}(t))^T]\mathbf{A}^T(\mathbf{y} - \mathbf{A}\hat{\mathbf{x}}_{0|t}(\mathbf{x}_t)) \tag{40}$$

$$= \mathbb{E}_{p_{0|t}(\mathbf{x}_t)}[f(X_0)^2] + \eta^2\|\mathbf{A}^T(\mathbf{y} - \mathbf{A}\hat{\mathbf{x}}_{0|t}(\mathbf{x}_t))\|_{\Sigma_{0|t}(\mathbf{x}_t)}^2 \tag{41}$$

$$- 2\eta\langle\mathbf{A}(\hat{\mathbf{x}}_{0|t,Y}(\mathbf{x}_t, \mathbf{y}) - \hat{\mathbf{x}}_{0|t}(\mathbf{x}_t)), \mathbf{y} - \mathbf{A}\hat{\mathbf{x}}_{0|t}(\mathbf{x}_t)\rangle. \tag{42}$$

Since $\mathbb{E}_{p_{0|t}(\mathbf{x}_t)}[f(X_0)^2] = \frac{\mathbb{E}_{p_{0|t}(\mathbf{x}_t)}[(p_{Y|0}(\mathbf{y}|X_0))^2]}{(\mathbb{E}_{p_{0|t}(\mathbf{x}_t)}[p_{Y|0}(\mathbf{y}|X_0)])^2} = 1 + \text{CV}^2(t)$, where $\text{CV}(t) = \frac{\sqrt{\text{Var}(p_{Y|0}(\mathbf{y}|X_0)|\mathbf{x}_t)}}{\mathbb{E}[p_{Y|0}(\mathbf{y}|X_0)|\mathbf{x}_t]}$ is the coefficient of variation of the likelihood function given $\mathbf{x}_t$.

---

**Algorithm 1** Inference-Time Search with Side Information for Inverse Problems

---

**Require:** Side information $\mathbf{s}$, observation $\mathbf{y}$, reward function $r$, resampling parameter $B$, number of particles $N$, temperature $\tau > 0$

1: Initialize $N$ particles: $\mathbf{x}_T[i] \sim \mathcal{N}(0, \mathbf{I})$ for $1 \leq i \leq N$
2: **for** $t = T - 1$ to $0$ **do**
3:      Sample $\mathbf{x}_t[i] \sim p_{t|t+1,Y}(\cdot \mid \mathbf{x}_{t+1}[i], \mathbf{y})$,               ▷ Sample candidate particles
4:      $r[i] \leftarrow r(\hat{\mathbf{x}}_{0|t,Y}[i]; \mathbf{s})$                 ▷ Compute reward using side information
5:      $g_t \leftarrow \text{GROUPSIZE}(N, B, t)$          ▷ Compute the group size at step $t$ for resampling
6:      $I \leftarrow \text{RESAMPLE}(r, g_t, \tau)$ ▷ Resample indices with replacement among the groups of size $g_t$
7:      $\mathbf{x}_t[i] \leftarrow \mathbf{x}_t[I[i]]$                  ▷ Retain the particles in the resampled indices
8: Select $\mathbf{x}_0^*$ from $\hat{\mathbf{x}}_{0|0}[1:N]$ (e.g., via reward maximization) **return** $\mathbf{x}_0^*$

---

### A.3 ALGORITHM

We summarize our framework in Algorithm 1. The GROUPSIZE step in line 5 computes the group-size at time $t$ and can be changed to obtain various search strategies: Best-of-N, Greedy Search, and RFJ Search, and the RESAMPLE step in line 6 samples the indices within the groups as described in the main paper. Line 3 is specific to the diffusion samplers and how they implement it.

Here, we roughly explain how the entire algorithm is implemented in all the three diffusion samplers.

**DPS:** Compute the denoised mean and the clean data estimate $\boldsymbol{\mu}_t[i], \hat{\mathbf{x}}_{0|t}[i]$ from $\mathbf{x}_{t+1}[i]$. Compute $\mathbf{g}_t[i] = \nabla_{\mathbf{x}_t} \|\mathbf{y} - \mathbf{A}\hat{\mathbf{x}}_{0|t}(\mathbf{x}_{t+1}[i])\|_2^2$ and use it to update $\hat{\mathbf{x}}_{0|t,Y}[i] \leftarrow \hat{\mathbf{x}}_{0|t}[i] - \frac{1-\alpha_t}{\sqrt{\alpha_t}}\eta\mathbf{g}_t[i]$, and $\boldsymbol{\mu}_t[i] \leftarrow \boldsymbol{\mu}_t[i] - \zeta\mathbf{g}_t[i]$. Then, compute rewards based on $\hat{\mathbf{x}}_{0|t,Y}[i]$ to resample promising indices. Take the reverse diffusion step on the resampled $\boldsymbol{\mu}_t[I[i]]$ to obtain $\mathbf{x}_t[i]$.

**DAPS:** Compute the clean data estimate $\hat{\mathbf{x}}_{0|t}[i]$ from $\mathbf{x}_{t+1}[i]$. Compute the rewards based on $\hat{\mathbf{x}}_{0|t}[i]$, resample, and then take MCMC steps, starting from the resampled particles, to perform a local Langevin sampling (Zhang et al., 2024). In the end, we obtain $\hat{\mathbf{x}}_{t,Y}[i]$, from which we sample $\mathbf{x}_t[i]$ by adding appropriate decoupled noise (Zhang et al., 2024).

**MPGD:** Compute the clean data estimate $\hat{\mathbf{x}}_{0|t}[i]$ from $\mathbf{x}_{t+1}[i]$. Compute gradient $\mathbf{g}_t = \nabla_{\mathbf{x}_{0|t}} \|\mathbf{y} - \mathbf{A}\hat{\mathbf{x}}_{0|t}[i]\|_2^2$. Take $\hat{\mathbf{x}}_{0|t,Y}[i] \leftarrow \hat{\mathbf{x}}_{0|t}[i] - \frac{1-\alpha_t}{\sqrt{\alpha_t}}\eta\mathbf{g}_t[i]$, and $\hat{\mathbf{x}}_{0|t}[i] \leftarrow \hat{\mathbf{x}}_{0|t}[i] - \zeta\mathbf{g}_t$. Resample indices based on rewards computed from $\hat{\mathbf{x}}_{0|t,Y}[i]$. Then, using the particles corresponding to the sampled indices $\hat{\mathbf{x}}_{0|t,Y}[I[i]]$, take a reverse DDIM step (He et al., 2024; Song et al., 2021a).

## B ADDITIONAL EXPERIMENTS

### B.1 PERCEPTUAL IMPROVEMENTS DO NOT GUARANTEE BETTER PSNR/LPIPS/SSIM

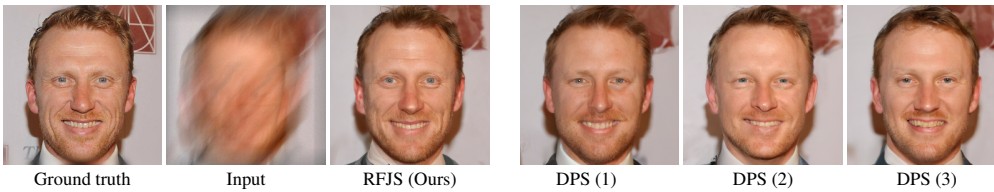

| Ground truth | Input | RFJS (Ours) | DPS (1) | DPS (2) | DPS (3) |

Figure 7: Qualitative illustration of the relevance of the FaceSimilarity metric and the superior performance of RFJS in identity preservation. RFJS reconstruction is clearly more faithful to the ground truth, yet PSNR, SSIM, and LPIPS values slightly favor the BlindDPS outputs.

Perceptual improvements do not necessarily lead to better values in classical metrics such as PSNR, LPIPS, and SSIM. Figure 7 provides a clear qualitative example: it compares three samples generated by BlindDPS with our RFJS reconstruction. Visually, our method preserves identity much better, and this is reflected in a lower FaceSimilarity. However, all three BlindDPS reconstructions achieve better PSNR, LPIPS, and SSIM than our result, showing that these traditional metrics can fail to capture semantic improvements.

We provide further evidence in Figure 8, where our reconstructions are perceptually closer to the ground truth than those of DPS, especially in challenging $32\times$ super-resolution settings. For these examples, Table 3 reports the standard metrics for our method and for reconstructions under DPS1. Once again, the perceptual gains visible in the images are not fully reflected in PSNR, LPIPS, or SSIM. This is important because the main goal in inverse problems is to recover the underlying *semantics* of the images, while PSNR, LPIPS, and SSIM are only proxies for reconstruction quality. When a metric better aligns with perceptual quality, it should be preferred. In our case, for face reconstruction this is captured by the FaceSimilarity metric, and for ImageNet settings by CLIPScore. In both cases, our method yields significant improvements over the baselines in these perceptual rewards.

This discrepancy is also intuitive from how these metrics are defined. PSNR is a purely pixel-level measure: it rewards putting the "right" colors in the "right" locations, even if the high-level content is not faithfully preserved. LPIPS and SSIM operate on features extracted by neural networks or on low-level structural patterns, and are more sensitive to capturing the correct class and coarse structure than to fine-grained semantics. For example, when reconstructing a human face, these metrics are not strongly incentivized to verify the correct identity; they mainly check that the output still looks like a plausible face. This explains why they may remain in the same range even when perceptual quality clearly improves.

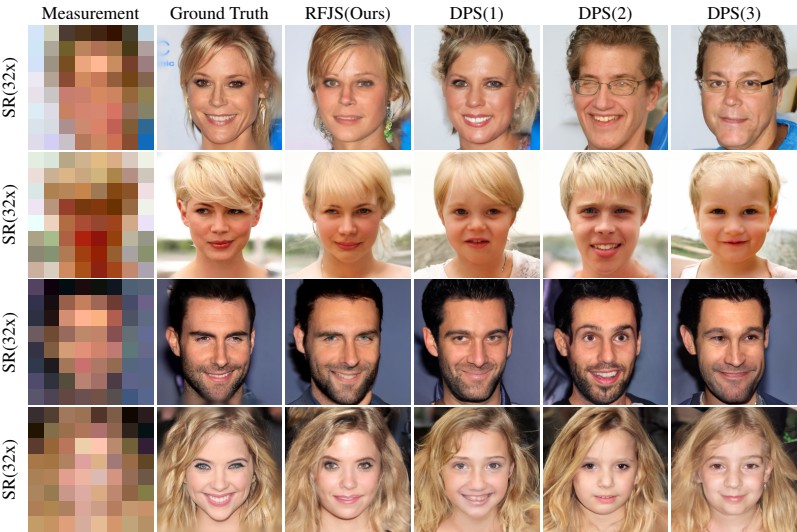

Figure 8: Our RFJS reconstructions preserve identity and perceptual details far better than DPS, as clearly visible from the images. This strong perceptual improvement is not reflected by classical metrics, which are reported in the table below.

| Image ID | PSNR | | LPIPS | | SSIM | |
|---|---|---|---|---|---|---|
| | RFJS | DPS1 | RFJS | DPS1 | RFJS | DPS1 |
| 1 | 20.30 | 19.98 | 0.363 | 0.353 | 0.549 | 0.541 |
| 2 | 19.12 | 18.77 | 0.382 | 0.413 | 0.567 | 0.552 |
| 3 | 20.95 | 20.98 | 0.254 | 0.273 | 0.609 | 0.596 |
| 4 | 17.85 | 18.54 | 0.411 | 0.406 | 0.368 | 0.389 |
| **Avg** | **19.56** | **19.57** | **0.353** | **0.361** | **0.523** | **0.520** |

Table 3: PSNR, LPIPS, and SSIM for the reconstructions shown in the figure above. Classical metrics fail to reflect the significant improvement.

## B.2 DPS

We also present additional qualitative examples for our DPS experiments in Figures 9 and 10, corresponding to settings that use face and text as side information, respectively.

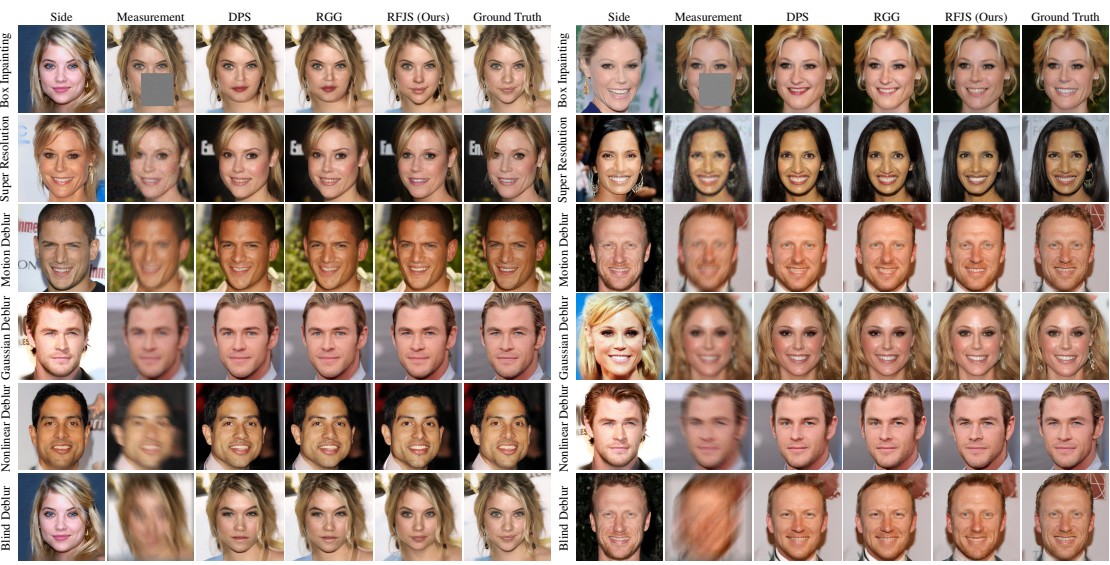

Figure 9: Additional samples using DPS as the base sampler.

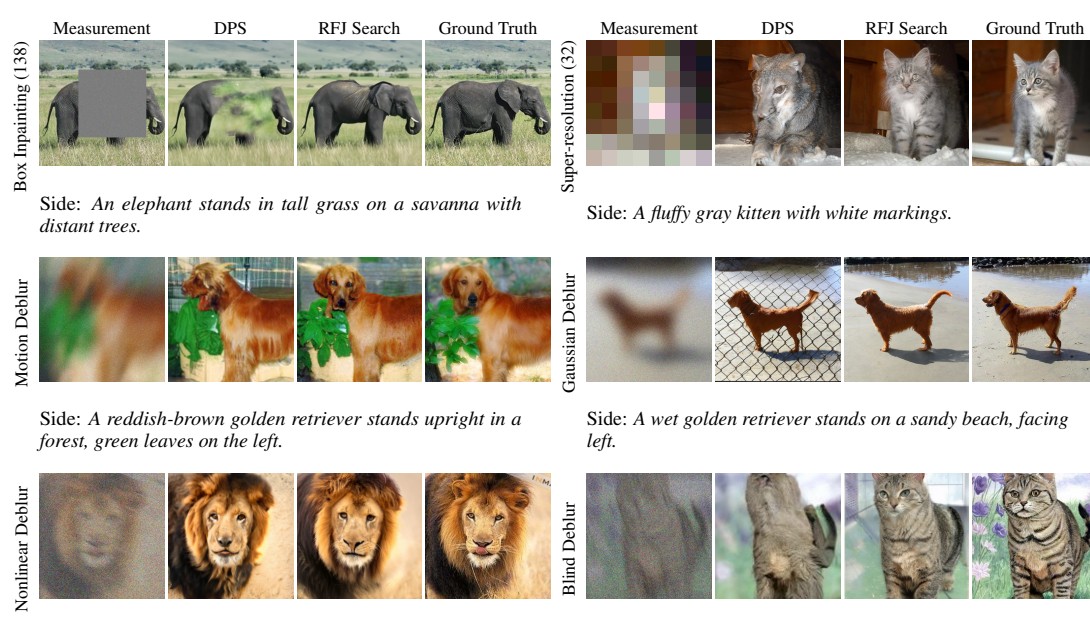

Figure 10: Qualitative comparison on ImageNet with textual side information. For highly degraded observations, DPS and BlindDPS often produce artifacts, whereas our method reduces these by better aligning reconstructions with the description.

## B.3 DAPS

**Setup:** We employ the search and gradient modules to infuse side information using DAPS as the base sampler. We consider two challenging tasks, box inpainting with a box of size $96 \times 96$ and

super-resolution with downsampling factor of 10. For gradient guidance, we used a scale of 15 with respect to the noise being added to $\hat{\mathbf{x}}_{0|t}(\mathbf{x}_t)$ after MCMC steps (Zhang et al., 2024). DAPS uses fewer diffusion steps (200) than DPS (1000). Further, the algorithm is based on a graphical model that allows for more inherent exploration due to the decoupling between consecutive steps. Therefore, for search algorithms, relatively smaller bases are preferable, and hence $B = 4$ is chosen.

**Results:** The qualitative results are given in Figure 11, while the quantitative metrics are given in 4. Observe that for the task of inpainting, our search algorithm shows a significant improvement over DAPS, particularly in the FaceSimilarity metric.

| | Box Inpainting | | | | Super Resolution ($\times 10$) | | | |
|---|---|---|---|---|---|---|---|---|
| Method | FaceSimilarity ($\downarrow$) | PSNR ($\uparrow$) | LPIPS ($\downarrow$) | SSIM ($\uparrow$) | FaceSimilarity ($\downarrow$) | PSNR ($\uparrow$) | LPIPS ($\downarrow$) | SSIM ($\uparrow$) |
| RFJS (ours) | **0.423**$_{\pm 0.10}$ | **28.720**$_{\pm 1.35}$ | **0.140**$_{\pm 0.03}$ | **0.788**$_{\pm 0.03}$ | $\underline{0.654}_{\pm 0.11}$ | $\underline{25.228}_{\pm 1.34}$ | **0.282**$_{\pm 0.03}$ | $\underline{0.661}_{\pm 0.04}$ |
| GS (ours) | 0.511$_{\pm 0.12}$ | 28.640$_{\pm 1.43}$ | 0.140$_{\pm 0.03}$ | 0.787$_{\pm 0.03}$ | 0.760$_{\pm 0.12}$ | **25.271**$_{\pm 1.36}$ | 0.285$_{\pm 0.03}$ | **0.662**$_{\pm 0.04}$ |
| RGG | $\underline{0.436}_{\pm 0.12}$ | 28.410$_{\pm 1.39}$ | 0.141$_{\pm 0.03}$ | 0.784$_{\pm 0.03}$ | **0.579**$_{\pm 0.13}$ | 25.210$_{\pm 1.34}$ | $\underline{0.282}_{\pm 0.03}$ | 0.659$_{\pm 0.04}$ |
| BON | 0.611$_{\pm 0.14}$ | $\underline{28.660}_{\pm 1.45}$ | 0.141$_{\pm 0.03}$ | 0.787$_{\pm 0.03}$ | 0.909$_{\pm 0.11}$ | 25.220$_{\pm 1.38}$ | 0.285$_{\pm 0.03}$ | 0.660$_{\pm 0.04}$ |
| DAPS | 0.739$_{\pm 0.18}$ | 28.290$_{\pm 1.53}$ | 0.142$_{\pm 0.03}$ | 0.784$_{\pm 0.03}$ | 1.020$_{\pm 0.14}$ | 25.170$_{\pm 1.35}$ | 0.285$_{\pm 0.03}$ | 0.659$_{\pm 0.04}$ |

Table 4: Comparison of metrics for various inverse problems using DAPS as the base sampler. For each metric, the best result is shown in **bold**, and the second best is underlined. We observe that our RFJ Search-based algorithm has the best or the second-best performance in all the tasks.

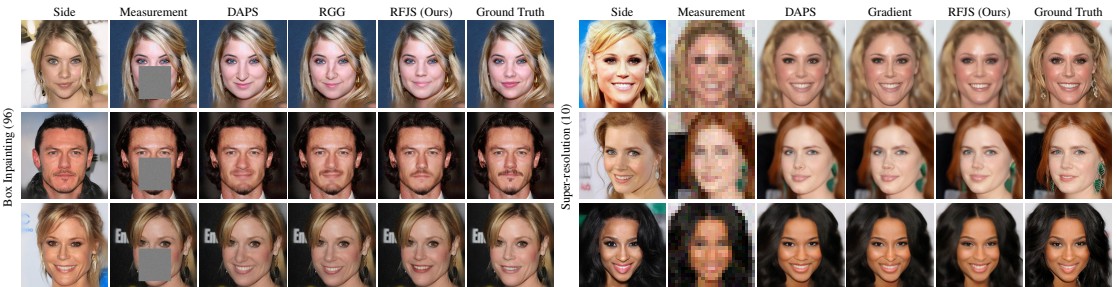

Figure 11: Qualitative comparison of algorithms using DAPS as the base sampler. Our method offers better reconstructions aligned with the identity.

### B.4 MPGD

**Setup:** As in the paper He et al. (2024), we choose super-resolution and Gaussian deblurring as the tasks, along with additional task of box inpainting. For box inpainting, we used a box of size $64 \times 64$ at the center of the face. Further, we modified the down-sampling scale of super-resolution from 4 to 6, and the intensity of the kernel in Gaussian deblur from 3 to 5 to make the tasks more challenging. Even though MPGD uses 100 DDIM steps in generation, its exploratory capabilities are similar to DPS. Therefore, we cannot use very large base $B$, whence, we choose $B = 8$ for box inpainting, and super-resolution. For Gaussian deblur, we found that using $B = 8$ becomes detrimental for other metrics, and so $B = 16$ is used. For the gradient guidance, the scales of $0.5$ for box inpainting, and $0.25$ for super-resolution and Gaussian deblur are chosen carefully to avoid overfitting. This is the scale relative to the gradient with respect to the measurement. For more discussion and examples on the effect of gradient scale, see Appendix B.6.

**Results:** Experimental results with MPGD (He et al., 2024) as baseline algorithm are given in Table 5. We observe that using RFJ Search significantly enhances the FaceSimilarity (FS) metric, while improving the other metrics. The reconstructions that utilize the side information exhibit strong identity match with the one described by the measurement, which is reflected in the FS metric.

| Sampler | Task | Method | FaceSimilarity (↓) | PSNR (↑) | LPIPS (↓) | SSIM (↑) |
|---|---|---|---|---|---|---|
| MPGD | Box Inpainting (64) | RFJS (8) (ours) | $\mathbf{0.542}_{\pm 0.08}$ | $\mathbf{29.81}_{\pm 1.44}$ | $\mathbf{0.102}_{\pm 0.02}$ | $\mathbf{0.852}_{\pm 0.02}$ |
| | | GS (8) (ours) | $\underline{0.587}_{\pm 0.10}$ | $\underline{29.44}_{\pm 1.75}$ | $\underline{0.102}_{\pm 0.02}$ | $\underline{0.851}_{\pm 0.02}$ |
| | | RGG (0.5) | $0.609_{\pm 0.08}$ | $29.24_{\pm 1.30}$ | $0.103_{\pm 0.02}$ | $0.850_{\pm 0.02}$ |
| | | BON | $0.661_{\pm 0.08}$ | $29.35_{\pm 1.82}$ | $0.102_{\pm 0.02}$ | $0.851_{\pm 0.02}$ |
| | | MPGD | $0.766_{\pm 0.07}$ | $29.09_{\pm 1.27}$ | $0.103_{\pm 0.02}$ | $0.848_{\pm 0.02}$ |
| | Super Resolution (6) | RFJS (8) (ours) | $\mathbf{0.834}_{\pm 0.09}$ | $\mathbf{24.50}_{\pm 1.48}$ | $\mathbf{0.242}_{\pm 0.04}$ | $\mathbf{0.666}_{\pm 0.06}$ |
| | | GS (8) (ours) | $0.878_{\pm 0.08}$ | $\underline{24.45}_{\pm 1.47}$ | $0.247_{\pm 0.03}$ | $0.660_{\pm 0.06}$ |
| | | RGG (0.25) | $\underline{0.854}_{\pm 0.07}$ | $24.39_{\pm 1.44}$ | $0.246_{\pm 0.03}$ | $0.656_{\pm 0.05}$ |
| | | BON | $0.964_{\pm 0.09}$ | $24.44_{\pm 1.58}$ | $\underline{0.244}_{\pm 0.04}$ | $\underline{0.664}_{\pm 0.06}$ |
| | | MPGD | $1.037_{\pm 0.07}$ | $24.39_{\pm 1.45}$ | $0.249_{\pm 0.03}$ | $0.657_{\pm 0.06}$ |
| | Gaussian Deblur (5) | RFJS (16) (ours) | $\underline{0.848}_{\pm 0.07}$ | $\underline{24.19}_{\pm 1.40}$ | $\mathbf{0.229}_{\pm 0.03}$ | $\underline{0.638}_{\pm 0.06}$ |
| | | GS (16) (ours) | $0.893_{\pm 0.07}$ | $24.14_{\pm 1.39}$ | $\underline{0.233}_{\pm 0.03}$ | $0.637_{\pm 0.06}$ |
| | | RGG (0.25) | $\mathbf{0.846}_{\pm 0.05}$ | $24.11_{\pm 1.34}$ | $0.235_{\pm 0.03}$ | $0.634_{\pm 0.05}$ |
| | | BON | $0.950_{\pm 0.07}$ | $\mathbf{24.20}_{\pm 1.38}$ | $0.233_{\pm 0.03}$ | $\mathbf{0.640}_{\pm 0.06}$ |
| | | MPGD | $1.026_{\pm 0.06}$ | $24.09_{\pm 1.35}$ | $0.236_{\pm 0.03}$ | $0.634_{\pm 0.06}$ |

Table 5: Quantitative comparison of reconstruction metrics in case of inverse problems with MPGD as the base sampler. For each metric, the best result is shown in **bold**, and the second best is underlined. Observe that our RFJSsearch algorithm has the best or the second-best performance in all the tasks. In Gaussian deblur, our search algorithm is only marginally worse than the best metrics attained. The value in the brackets indicates the resampling rate for search algorithms, and gradient scale for Gradient algorithm.

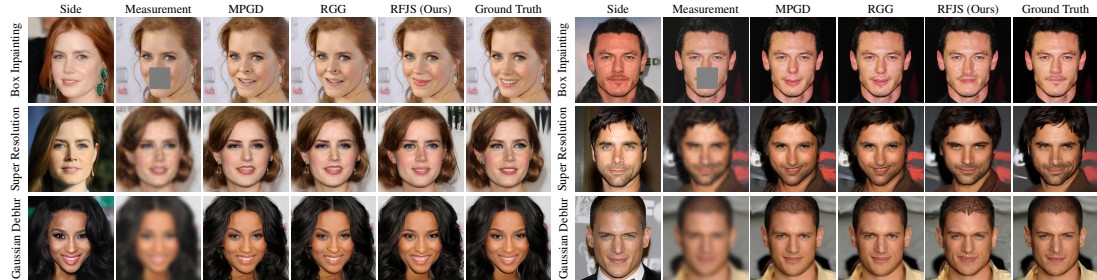

Figure 12: Qualitative comparison of algorithms using MPGD as the base sampler. Our method offers better reconstructions aligned with the identity. Notice that while the identity is preserved, the exact reconstruction might not be possible as witnessed in, for example, Box inpainting. The ground truth has a smiling face whereas the reconstruction does not, although being the same person. Thus, PSNR improvements over the base sampler might be small, but FaceSimilarity improvements are significant.

## B.5 ABLATION STUDIES

In this section, we conduct ablation studies to study the performance and robustness of our algorithm by degrading: (i) Measurement quality and (ii) Side information quality.

### B.5.1 MEASUREMENT QUALITY: HARD INVERSE PROBLEMS

In this section, we demonstrate the effectiveness of our method on hard inverse problems with severely degraded measurements. For DPS, we evaluate inpainting with a large mask that covers almost the entire face, super-resolution with aggressive downsampling ratios of 12× and 32×, and motion deblurring using a large 256-pixel kernel. For MPGD, we similarly consider challenging variants of inpainting, super-resolution, and Gaussian deblurring, with quantitative results reported in Tables 6 and 7. Across all settings, our method consistently improves the target reward metric, which translates directly into stronger qualitative performance, and we also observe improvements in classical metrics on average. The qualitative results are shown in Figures 13, 14, 8. For textual side information, we already include challenging experimental settings in the main paper; see Figure 4 and Table 2.

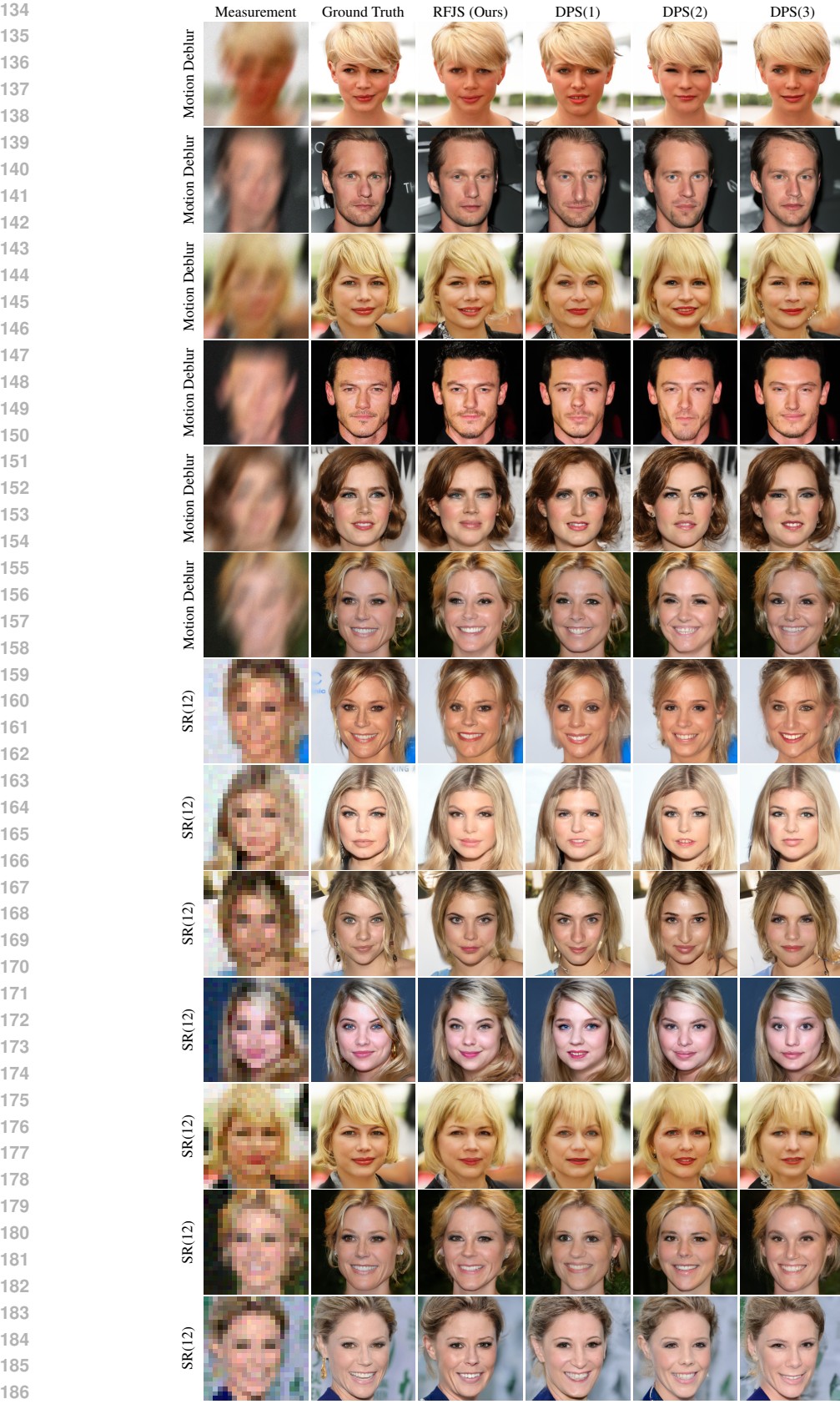

Figure 13: Qualitative results on hard tasks using DPS. Our algorithms reconstructs faces with perceptual quality.

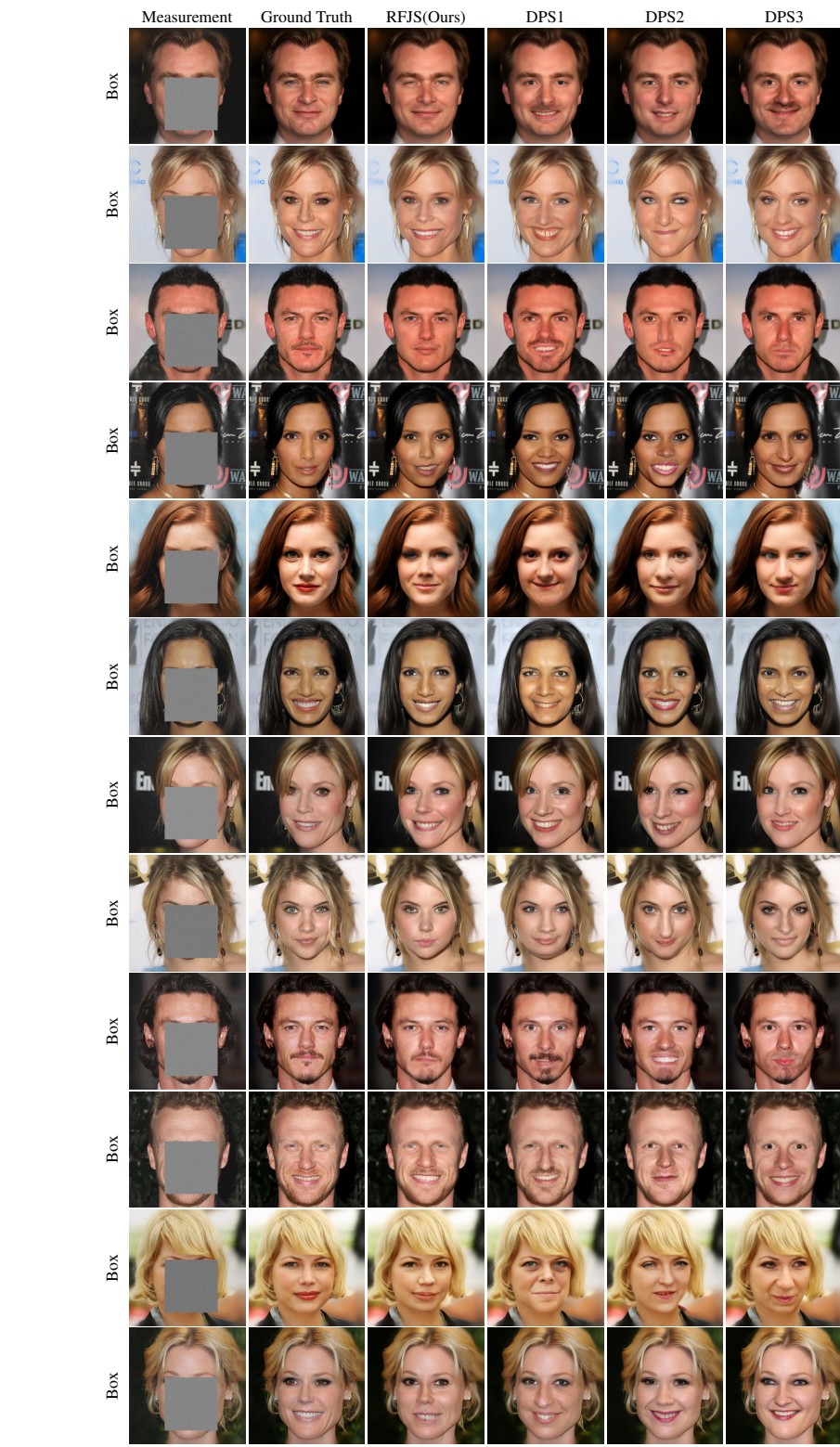

Figure 14: Qualitative results on box inpainting using DPS sampler. Since the face is completely masked in the measurement, side information provides useful clues so that the reconstruction preserves identity.

| Task Name | Task Parameters | Algo | FS (↓) | PSNR (↑) | LPIPS (↓) | SSIM (↑) |
|---|---|---|---|---|---|---|
| Box Inpainting | $M = 116$ | RFJS | **0.47** | **26.62** | **0.140** | **0.839** |
| | | DPS | 1.07 | 26.21 | 0.151 | 0.826 |
| Super Resolution | $S = 12$ | RFJS | **0.603** | **22.92** | **0.262** | **0.619** |
| | | DPS | 1.23 | 22.78 | 0.265 | 0.613 |
| | $S = 32$ | RFJS | **0.748** | **18.66** | **0.354** | **0.496** |
| | | DPS | 1.38 | 18.43 | 0.363 | 0.489 |
| Motion Deblur | $K = 256$ | RFJS | **0.545** | 22.67 | **0.257** | 0.614 |
| | | DPS | 1.21 | **22.69** | 0.260 | **0.615** |

Table 6: Quantitative results using DPS for hard tasks with dps scale $= 0.8$ (guidance strength from the measurement $y$)

| Task Name | Task Parameters | Algo | FS (↓) | PSNR (↑) | LPIPS (↓) | SSIM (↑) |
|---|---|---|---|---|---|---|
| Inpainting | $M = 96$ | RFJS | **0.594** | **24.83** | **0.171** | **0.749** |
| | | MPGD | 0.831 | 24.79 | 0.172 | 0.747 |
| Super Resolution | $S = 12$ | RFJS | **1.032** | **21.40** | **0.312** | **0.555** |
| | | MPGD | 1.282 | 21.32 | 0.318 | 0.553 |
| | $S = 16$ | RFJS | **1.142** | **19.03** | 0.383 | **0.486** |
| | | MPGD | 1.356 | 19.01 | **0.381** | 0.486 |
| Gaussian Deblur | $K = 81, I = 5.0$ | RFJS | **0.820** | **24.49** | **0.238** | **0.645** |
| | | MPGD | 0.993 | 24.39 | 0.241 | 0.641 |

Table 7: Quantitative results using MPGD for hard tasks with scale $= 6.0$ (guidance strength from the measurement $y$)

### B.5.2 SIDE INFORMATION QUALITY

In this subsection, we evaluate how the quality of side information affects the performance of our method. For the face experiments, we study a $4\times$ super-resolution task using DPS as the base sampler. We degrade the side information by applying a Gaussian blur with a $31 \times 31$ kernel and varying blur intensities. Figure 15 reports the resulting Face Similarity scores. As expected, performance consistently improves as the side information becomes more reliable. The case with blur intensity 0.0 corresponds to the unaltered side information and matches the results reported in the main paper. Representative qualitative examples are provided in Figure 16. Notably, even with heavily degraded side information (intensity 5.0), our method (FS: 0.53) outperforms the baseline (FS: 1.042), and the performance steadily improves as the fidelity of the side information increases.

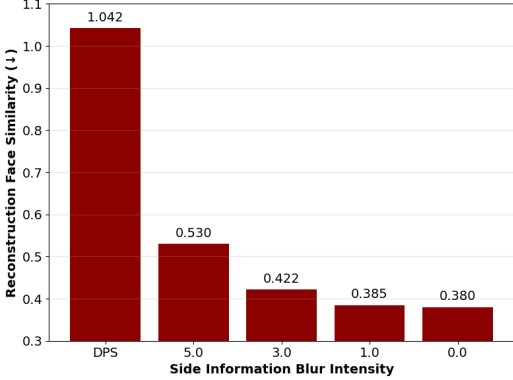

Figure 15: Performance of RFJS algorithm with side information blurred with Gaussian kernel in the face data experiments

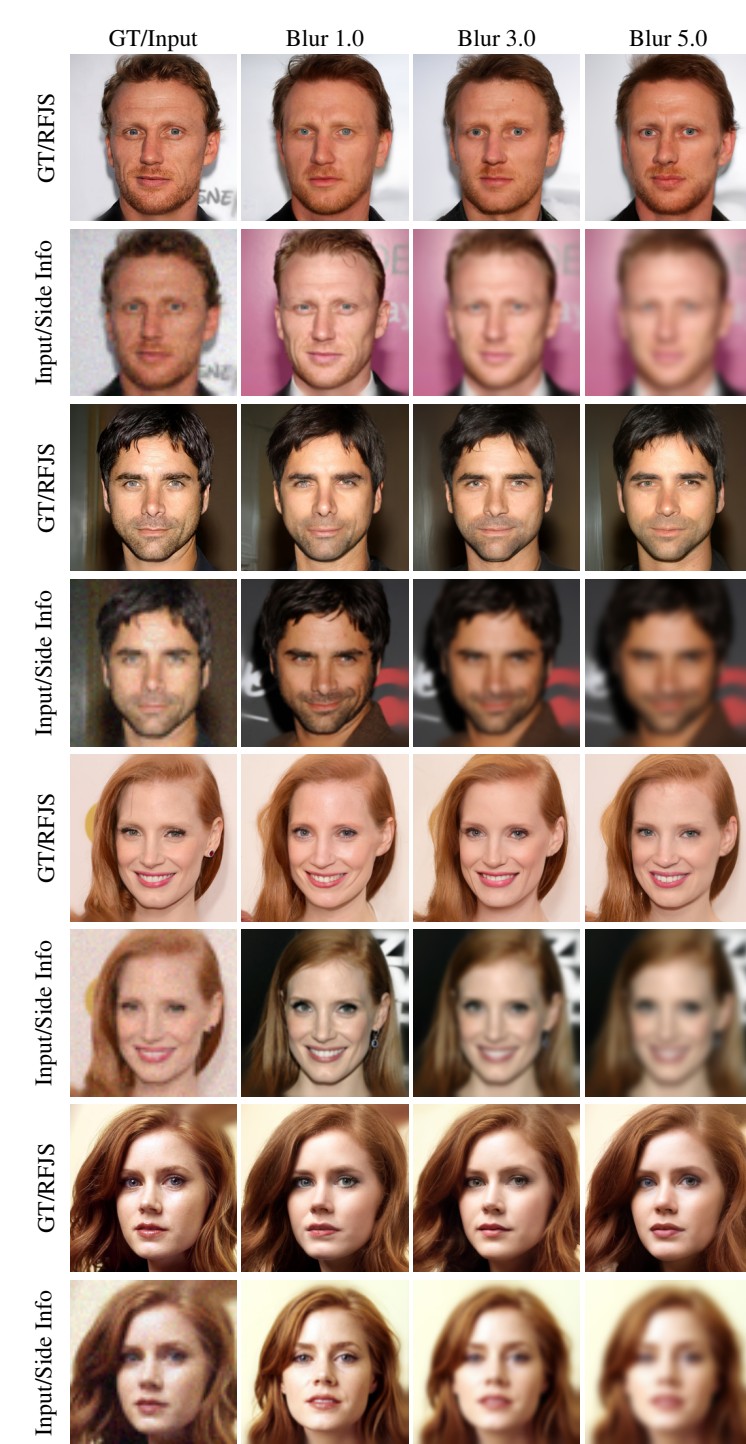

Figure 16: Qualitative comparison of generations with side information blurred at different levels. Some sharp features that preserve identity are lost when the side information is severely blurred.

For the case of textual side information, Figure 17 shows that our method remains consistent under small variations in the prompt. In this example, the key missing information in the measurement is the type of animal or object present in the scene. Once the side information specifies that there is a golden retriever, the reconstruction improves significantly. Further modifying the prompt does not

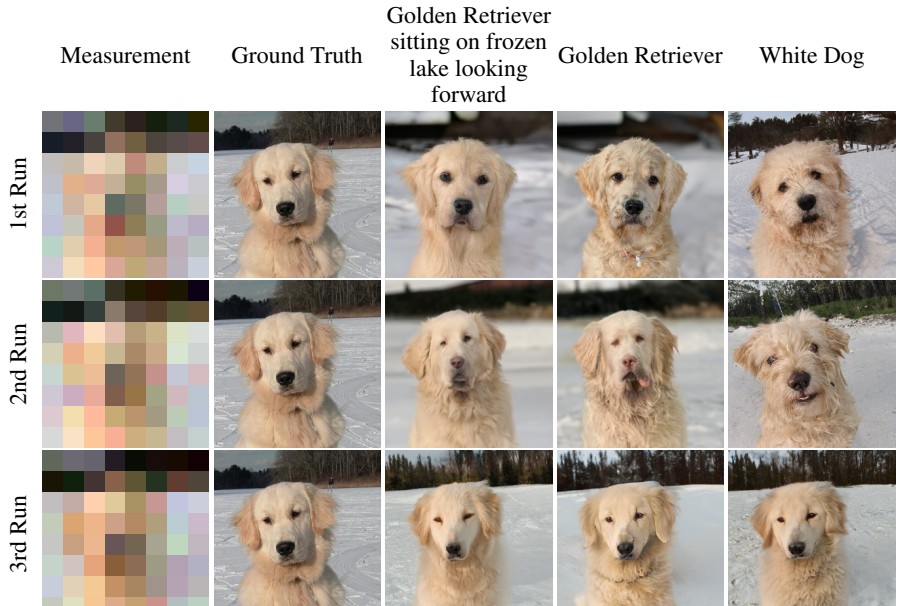

Figure 17: Qualitative comparison of generations with different textual descriptions as side information. Due to randomness and robustness of the algorithm due to relative ordering, the quality assessment is difficult.

substantially affect the output. If the side information is "white dog", then our algorithm reconstructs a dog which is not necessarily a golden retriever (Figure 17, last column).

### B.6 EXPERIMENTS ON EFFECT OF GRADIENT SCALE

**Gradient Guidance Limitations.** While guiding the reverse diffusion process with reward gradients can help generate images with higher reward scores, this approach has several limitations. First, as shown in Figures 18, 19, gradient-based guidance primarily adds fine details, such as wrinkles or texture, to the reconstruction, but it cannot significantly alter the global structure of the face. To isolate the effect of the gradient, we used fixed noise realizations for both the gradient-based and baseline methods. The results show that changes are mostly confined to local details, implying that if the sampling trajectory is poor, gradient guidance alone cannot compensate. This highlights the need for search-based methods that can explore a wider range of trajectories during inference.

Second, this method is sensitive to the choice of gradient scale. In the visual examples, we used a relatively large scale of 1.6 to make the gradient's effect more visible; however, such high scales often degrade other metrics like PSNR and SSIM and introduce artifacts. Empirically, we found that a scale around 0.5 yields the best balance when the base sampler is DPS or BlindDPS, consistently improving the FaceSimilarity metric while preserving other evaluation metrics and avoiding artifacts (see Figure 19). Moreover, the sensitivity to gradient scale increases when the number of reverse diffusion steps is small. For instance, in DAPS (Figure 18) and MPGD, where the number of steps is limited to 200 and 100 respectively, larger scales quickly lead to visible artifacts, as demonstrated in Figures 20, and 21.

It is well established in deep learning research that deep (convolutional) neural networks are vulnerable to gradient-based adversarial attacks (Goodfellow et al., 2015). Consequently, using such networks to provide reward-based guidance inherits these vulnerabilities. However, when combined with diffusion samplers, this susceptibility is partially alleviated, as the diffusion process can help steer trajectories away from adversarially induced local minima. This mitigating effect is particularly evident when using a large number of sampling steps (e.g., 1000 steps in DPS). In contrast, samplers with fewer steps (e.g., 100 steps in MPGD) exhibit increased sensitivity to the gradient scale, as illustrated in Figures 20 and 21.

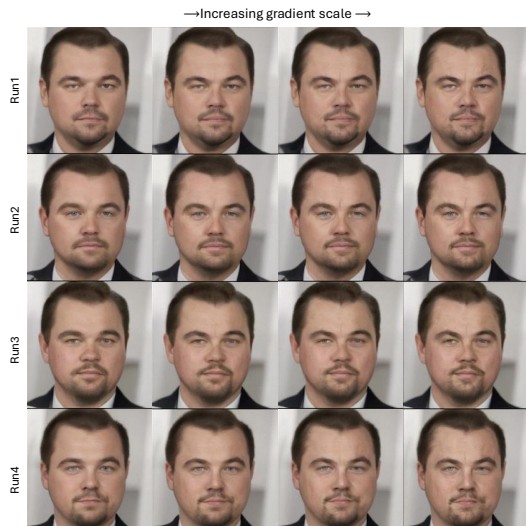

Figure 18: **Effect of reward-gradient guidance in diffusion-based inverse problems.** We show 4 runs with different random seeds (rows), and for each seed we vary the gradient scale across 4 settings (columns). Within each row, the noise realization is identical and only the gradient scale changes; within each column, the gradient scale is fixed while the random seed varies. The ground truth and degraded input are the same for all reconstructions. This arrangement reveals two key observations: (1) The reward gradient influences fine details, such as wrinkles and facial lines without altering the overall facial structure; the structure is primarily determined by the initial noise realization. (2) Different seeds reconstruct different face structures, highlighting the multi-modal nature of the problem. This demonstrates why using multiple particles and performing search across them is beneficial: it enables exploration of structurally different hypotheses while the reward gradient refines locally.

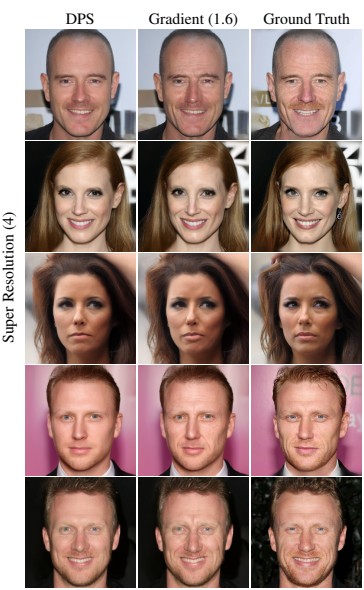

Figure 19: Qualitative comparison of the effect of gradient scale on reconstruction paths.

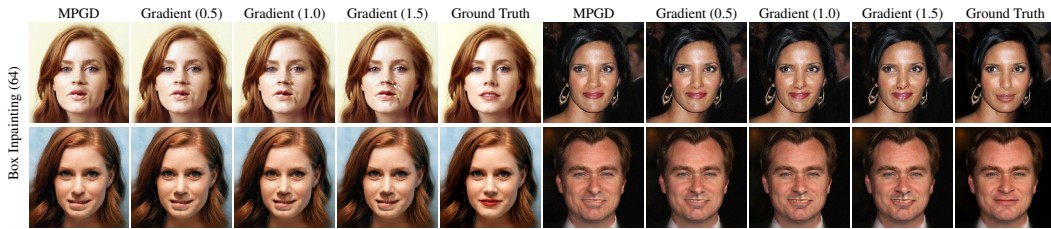

Figure 20: Qualitative comparison of the effect of gradient scale on reconstruction paths. Notice that while the base reconstruction is reasonable, adding the gradient can degrade it if the scale is very large. The final scale used in the experiments is $0.5$.

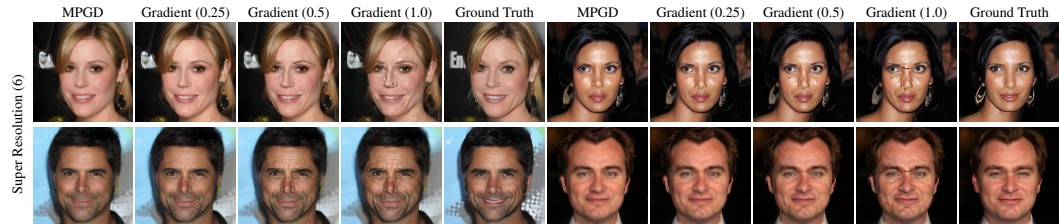

Figure 21: Qualitative comparison of the effect of gradient scale on reconstruction paths for super resolution task. Notice that while the base reconstruction is reasonable, adding the gradient can degrade it if the scale is very large. The final scale used in the experiments is $0.25$.

### B.7 EFFECT OF NUMBER OF PARTICLES

In this section, we study how performance scales with the number of particles. Figure 22 presents the results for the box inpainting task using RFJS with DPS as the baseline sampler, while Figure 23 reports the corresponding results for the remaining tasks. The evaluations using MPGD as the baseline are shown in Figure 24. Across all tasks and for both baseline samplers, we observe a consistent improvement in performance as the number of particles $N$ increases, aligning with the expected behavior of particle-based search methods. Furthermore, our results indicate that RFJS scales more efficiently with increasing $N$ than both Greedy Search and Best-of-N.

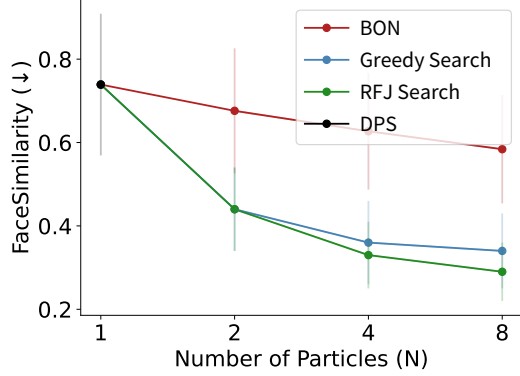

Figure 22: Scaling of search algorithms with respect to the number of particles.

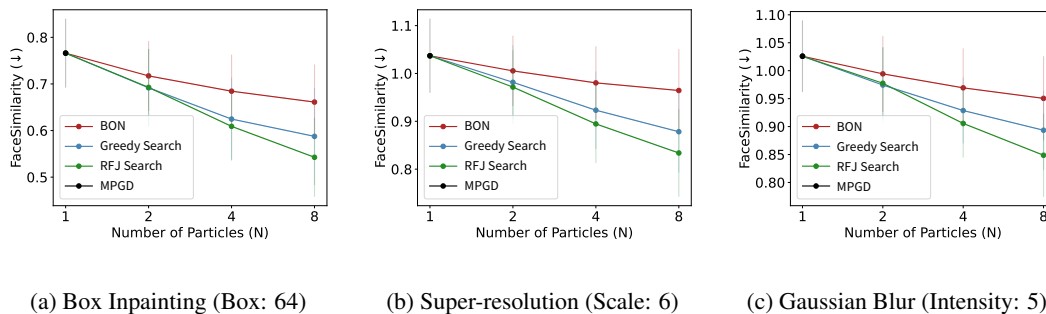

(a) Box Inpainting (Box: 64)     (b) Super-resolution (Scale: 6)     (c) Gaussian Blur (Intensity: 5)

Figure 24: Effect of number of particles $N$ on the FaceSimilarity metric. RFJ Search algorithm offers the best scaling performance, followed by Greedy Search algorithm. Finally, BestOfN performance improves, but only marginally. $64$ indicates the size of the box for inpainting, $6$ indicates the down-sampling factor in super-resolution, and $5$ is the intensity of the Gaussian kernel in Gaussian deblur.

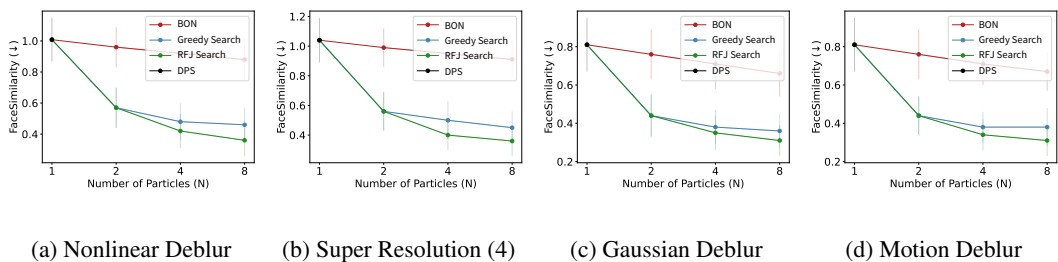

(a) Nonlinear Deblur     (b) Super Resolution (4)     (c) Gaussian Deblur     (d) Motion Deblur

Figure 23: Effect of the number of particles $N$ on the FaceSimilarity metric in DPS. As $N$ increases, the performance improves.

## B.8 RUNTIMES

Table 8 reports wall-clock runtimes (in seconds) for our search algorithms compared to Best-of-N (BON) and Greedy Search across different baselines (DPS, DAPS, and MPGD) with varying numbers of particles. When the number of particles is set to $1$, the runtime corresponds to the baseline method without search. As the number of particles increases, amount of computation scales linearly with $N$, but thanks to parallelization, the wall-clock overhead remains moderate: with $N = 8$, runtimes are only $4 - 5\times$ those of the baseline. We also note that our RFJ Search method take slightly more time than BON and Greedy Search, but consistently achieve better reconstruction quality, highlighting the practical efficiency of our approach.

Finally, resampling of the particles requires computing the rewards, which involves call to the reward evaluation model. Therefore, decreasing the base $B$ (more frequent resampling) increases the wall-clock time. Denoting $c_d, c_r$ the computation cost per function call to diffusion model and reward model respectively, a rough estimate of the time taken to run the algorithm is given by:

$$N(O(Tc_d) + O((T/B)c_r)) \leq N(O(Tc_d) + O(Tc_r)),$$

where $T$ denotes the number of diffusion steps and $N$ the number of particles. However, it is important to note that the wall-clock time is dominated by the call to the diffusion model, which is a much larger network than the reward model, i.e, $c_d \gg c_r$ in practice, and so the reward evaluation does not add a significant overhead.

| Particles | BON | Greedy Search | RFJ Search |
|---|---|---|---|
| 1 | 55 | - | - |
| 2 | 65 | 75 | 75 |
| 4 | 102 | 118 | 131 |
| 8 | 180 | 195 | 241 |

(a) Task: Box inpainting. Baseline: DPS

| Particles | BON | Greedy Search | RFJS |
|---|---|---|---|
| 1 | 61 | - | - |
| 2 | 72 | 91 | 91 |
| 4 | 125 | 141 | 157 |
| 8 | 229 | 245 | 290 |

(b) Task: Box inpainting. Baseline: DAPS

| Particles | BON | Greedy Search | RFJS |
|---|---|---|---|
| 1 | 3 | - | - |
| 2 | 4 | 5 | 5 |
| 4 | 5 | 8 | 9 |
| 8 | 8 | 12 | 23 |

(c) Task: Box inpainting. Baseline: MPGD

Table 8: Runtime (seconds) vs number of particles for BON, Greedy Search, and RFJ Search methods on DPS, DAPS and MPGD ($B = 8$). The baseline algorithm corresponds to $N = 1$.

### B.9 Evaluating Search Strategies in a 2D Setting

To illustrate the effect of side information in inference-time search, we consider a simple 2D setup. The prior on $x_0$ is a mixture of Gaussians, shown in the leftmost image of Figure 26. A ground-truth sample $x_0$ is drawn from this prior, and a random forward operator $A$ is generated with fixed norm and random orientation. The observation is then $y = Ax_0 + n$. The corresponding posterior $p(x_0 \mid y)$, approximated by DPS, is also shown in Figure 26. Because this is an ill-posed inverse problem, the posterior is multimodal and DPS fails to reconstruct the correct solution.

We then add side information of the form $s = Dx_0 + n_s$, where $D$ is chosen so that $Dx_0$ is orthogonal to $Ax_0$. The reward function is defined as $r(x_t, s) = -\|s - D\hat{x}_0(x_t)\|_2^2$, where $\hat{x}_0(x_t)$ is the Tweedie estimate of the clean signal from the noisy state $x_t$. Using this reward, RFJ Search produces posterior samples $p(x_0 \mid y, s)$ as shown in the three rightmost images of Figure 26. As the resampling base $B$ decreases, the particles concentrate more tightly around the ground truth, highlighting how side information guides inference.

To compare RFJ and Greedy Search, we next consider a more realistic scenario where side information is generated through a neural network reward model. For example, in face reconstruction tasks, side information may come from the embedding of an additional image, while for text-conditioned tasks, it may come from a text encoder. To mimic this, in each trial we sample a ground truth $x_0$, a forward operator $A$, and a reward network $r_\theta$ with randomly initialized weights. The side information is defined as $s = r_\theta(x_0) + n_s$, and the reward is the cosine similarity between $r_\theta(\hat{x}_0(x_t))$ and $s$. We repeat this experiment 16,000 times, and for each trial run both RFJ and Greedy Search 128 times with $N = 8$ particles, varying the resampling base $B$ over powers of two from 1 to 256. The average PSNR values are reported in Figure 25. Across all settings, RFJ Search consistently outperforms Greedy Search, with the best performance observed at $B = 4$.

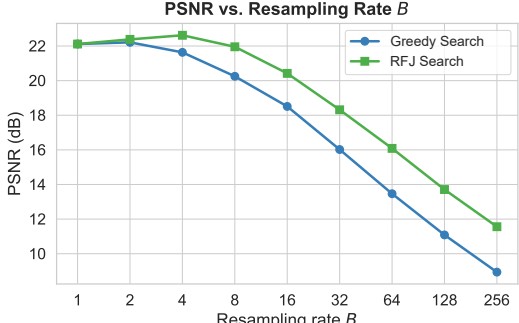

Figure 25: Comparison of performance of RFJ Search (RFJS) and Greedy Search (GS) as a function of $B$ for a randomly generated reward network $r_\theta$. RFJS outperforms GS across all values of $B$.

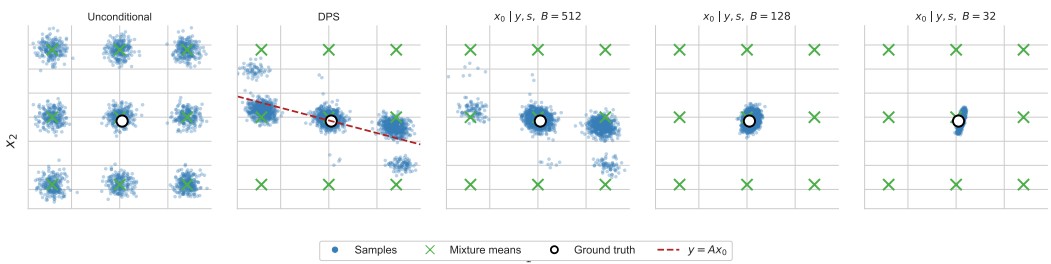

Figure 26: Illustration of the effect of $B$ in utilization of side information for the reconstruction in a linear inverse problem with a mixture of Gaussian prior.

## C  HYPERPARAMETERS

In our implementation of RESAMPLE in Algorithm 1, we perform a greedy resampling, i.e., we pick the best candidate within each group of size $g_t$ and replicate it $g_t$ times. Since we are using large enough $B$, this is justified and has similar effect as using smaller $B$ with moderate temperature, with the added advantage of utilizing less function calls to the reward network $r$. Thus, tuning the $B$ allows us to maintain balance without over-optimizing with respect to the reward.

For reproducibility, we provide detailed settings for each task, sampler, and search algorithm used in our experiments below.

### C.1  FACE IDENTITY EXPERIMENTS

For experiments with face identity as side information:

- **DPS:** Box inpainting with box size 96. All other parameters (downsampling rate, blur kernel, noise levels) are the same as the default DPS settings in the implementation. Search algorithms use $N = 8$ particles and resampling base $B = 16$ for both RFJ and Greedy search. Gradient guidance is applied with scale 0.5.

- **DAPS:** Box inpainting with box size 96, super-resolution with downsampling rate $10\times$. Noise levels unchanged from original DAPS defaults. Search algorithms use $N = 8$ particles and resampling base $B = 4$, with gradient guidance scale 13.

- **MPGD:** Box inpainting with box size 64, super-resolution with downsampling rate $6\times$, and Gaussian deblur with intensity of 5.0. Search algorithms use $N = 8$ particles and resampling base $B = 8$. Gradient guidance scale is 0.5 for box inpainting and 0.25 for super-resolution and Gaussian deblur.

### C.2  TEXT SIDE INFORMATION EXPERIMENTS

When text descriptions were used as side information, we made the degradation more severe so that the information in $\mathbf{s}$ was not already present in the measurement $\mathbf{y}$. Otherwise, side information would not provide meaningful guidance. For example, if the input image is sharp enough to identify the type of animal, then explicitly stating it in $\mathbf{s}$ adds little value.

The settings for these tasks are:

- **Box inpainting:** Box size 138, noise level same as default.

- **Super-resolution:** Downsampling rate $32\times$, noise level same as default.

- **Motion/Gaussian deblur:** Kernel size 256, intensity of Gaussian 5.0, noise level 0.1.

- **Nonlinear/Blind deblur:** Kernel size unchanged, noise level 0.5.

For all tasks in this setting, search algorithms use $N = 4$ particles and resampling base $B = 100$.

These hyperparameters ensure that our framework is evaluated under severe degradations (heavy downsampling, blurring, or noise), while search and guidance settings remain consistent across samplers and modalities.

### C.3  MRI EXPERIMENTS

We used the contrast-pairings among the files in the fastMRI dataset, provided by Atalık et al. (2025). We collect the data from the (fastMRI) source and preprocess to be compatible with the inputs in ContextMRI. Specifically, the setup used in the data is multi-coil MRI acquisition, which requires us to estimate the coil sensitivity maps, and then a complex reconstruction from them. ContextMRI takes complex values as inputs and denoises to produce a complex-valued 2D image. We computed NMI with 64 bins at each step of the diffusion process to balance complexity with performance. We use the defaults parameters as in ContextMRI, except for the acceleration factor, 16 and the center fraction (ACS), 0.02. We use a pair of anatomy which two contrasts, which has more 30 slices. We consider the slices 15 to 28 as these are more challenging and report the results by using one as the side information for the other.

## D  LIMITATIONS

Our proposed search algorithms lack formal optimality guarantees for exploration–exploitation, and we do not claim they are theoretically optimal. We expect that stronger algorithms are possible, potentially improving both sample efficiency and robustness. A central reason is the absence of a general mathematical framework for designing optimal exploration–exploitation strategies in diffusion-based inverse problems with side information, an open problem we highlight. Practically, this means our methods rely on principled heuristics and tuned schedules (e.g., reward scaling, resampling/branching rates) whose compute allocation is not provably optimal, suggesting a clear direction for future work.

## E  USE OF LARGE LANGUAGE MODELS

Parts of this work were assisted by large language models (specifically GPT-5 from OpenAI). Their use was limited to improving clarity, grammar, and the presentation of experimental descriptions. All conceptual contributions, experimental design, analysis, and final decisions are solely the authors' responsibility. The models were not used to generate new research ideas, design experiments, or make unverifiable scientific claims. Additionally, large language models were used to generate textual descriptions serving as side information for a specific set of experiments, as detailed in the main paper.