# OpenReview forum: "Inference-Time Search using Side Information for Diffusion-based Image Reconstruction"
_ICLR.cc/2026/Conference — Submitted to ICLR 2026_

### Official Review · Reviewer_kQJ7 · 2025-10-31

**Soundness:** 4
**Presentation:** 4
**Contribution:** 3
**Rating:** 6
**Confidence:** 4

**Summary:**

The paper proposed a method to take advantage of side information in solving ill-posed inverse problems with diffusion prior. Side information is incorporated by a reward function and is integrated in a plug-and-play manner into any inverse problem solvers using pre-trained diffusion model, such as DPS and DAPS. Instead of using reward gradient guidance, the paper adopts inference-time search techniques to accommodate general reward functions. Extensive experiments show superior performance comparing to baselines across various inverse problems on a number of base models.

**Strengths:**

1. Novelty: The paper is the first to incorporate side information for diffusion inverse problems.
2. Flexibility: The proposed method integrates smoothly into a wide range of diffusion inverse problem solvers, and is compatible with general reward functions.
3. Effectiveness: Experimental results suggest that the method outperforms baselines under various scenarios.

**Weaknesses:**

1. Although experiments show that the proposed method significantly outperforms baseline methods without side information, it is not clear how side information quality could affect the reconstruction.
2. Since highly informative side information is provided, the authors should consider testing on "difficult" scenarios such as high measurement noise and severe ill-posedness.
3. The method involves various approximations such that theoretical insights are intractable.

**Questions:**

1. Is it possible to compare reconstruction quality with side information with different quality? For example, in a fixed inverse problem, it could be interesting to see the difference between text "golden retriever sitting on a snowy frozen lake, facing forward", "golden retriever sitting on a snowy frozen lake", "golden retriever sitting", and "dog". It could be also interesting to see how a blurry image (or a side view when the ground truth is a front view) side information could affect the reconstruction.
2. Can the authors apply the algorithm on more challenging inverse problems as mentioned in the second point of Weaknesses?
3. Why does DPS fail completely in the results of Figure 4?

---

> ### Author Response · Authors · 2025-11-21
> **Author Response**
>
> We thank the reviewer for the positive comments regarding the novelty, flexibility, and effectiveness of our approach. Based on the reviewers’ suggestions, **we have now included additional experimental results and discussions in Appendix B.1 and B.5**. We hope that these additions will further clarify the novelty and effectiveness of our contributions. Below, we address all the questions and comments raised by the reviewer. We have also updated the **supplementary material** to incorporate these revisions. For the reviewers' convenience, all modifications relative to the original submission are highlighted in blue.
>
> ---
>
> **Q1.**
> “Is it possible to compare reconstruction quality with side information with different quality?” and “… it is not clear how side information quality could affect the reconstruction.”
>
> **Response:**
> In our revised version, **we have included additional experimental results in Appendix B.5.2** to address this concern, by evaluating how the side information quality affects the performance of our algorithm. We consider 4× super-resolution using DPS as the baseline sampler for face data. We evaluate the performance of our RFJS algorithm with guidance provided by side information pictures blurred by a Gaussian kernel of varying intensities (1, 3, 5); see Figures 15 and 16 in the revision. We observe that even with degraded (intensity 5) side information, our algorithm (FS: 0.53) improves over the baseline (FS: 1.042) without any side information, and the performance improves as the quality of the side information improves.
>
> For the exact experiment on textual side information suggested by the reviewer (Question 1), see Figure 17 of Appendix B.5.2. In this example, the key missing information in the measurement is the type of animal or object present in the scene. Once the side information specifies that there is a golden retriever, the reconstruction improves significantly. Further modifying the prompt does not substantially affect the output. If the side information is “white dog”, then our algorithm reconstructs a dog which is not necessarily a golden retriever.
>
> ---
>
> **Q2.**
> “… consider testing on ‘difficult’ scenarios such as high measurement noise and severe ill-posedness.”
>
> **Response:**
> **We have now conducted the following additional experiments on difficult tasks:**
> (i) super-resolution (12×, 16×, 32×),
> (ii) box inpainting (96×96, 116×116),
> (iii) motion deblur (kernel size 256×256),
> (iv) Gaussian deblur (kernel size 81×81).
>
> The qualitative results are shown in Figures 13 and 14, and the quantitative results are shown in Tables 6 and 7, **included in a new subsection, Appendix B.5.1**. These extensive experiments further reinforce our central claim that the proposed method yields consistently higher-quality reconstructions across all evaluated tasks. This is particularly evident in the substantial gains observed in Face Similarity, highlighting the effectiveness of our approach.
>
> ---
>
> **Q3.**
> “The method involves various approximations such that theoretical insights are intractable.”
>
> **Response:**
> The goal of the theoretical results is to provide insight about the consistency of our approximations and to enable sampling from the approximate posterior $p_{X \mid Y, S}(x \mid y, s)$. We show (in Proposition 3) that the **approximation error is small** when $t$ is small. Effectively, since the distribution $p_{0 \mid t}(x_0 \mid x_t)$ concentrates around $x_t$ as $t$ becomes small, the variations $(c_1(t), c_2(t), c_3(t), c_4^\eta(t))$ become small, and the approximation error stays small (Remark 4).
>
> Empirically, we show that we are indeed sampling from a tilted posterior, which is reflected in qualitative and quantitative improvements (particularly, the target metric of Face Similarity). The theory is meant to strengthen the claim that (under some assumptions) our approximations are valid and consistent.
>
> ---
>
> **Q4.**
> (Question 3) “Why does DPS fail completely in the results of Figure 4?”
>
> **Response:**
> DPS fails in Figure 4 because, as seen in the “Measurement” column, the inverse problems are highly ill-posed: large box inpainting (box: 136), super-resolution (32×), high blur (kernel size 256, intensity 5.0), high noise ($\sigma = 0.1, 0.5$). Refer to Appendix C.2 for exact details.

---

### Official Review · Reviewer_1jpL · 2025-10-31

**Soundness:** 3
**Presentation:** 3
**Contribution:** 3
**Rating:** 6
**Confidence:** 4

**Summary:**

This paper introduces a training-free inference-time search framework that leverages side information to guide diffusion-based image reconstruction. By modeling side information as a reward-tilted prior and applying greedy or recursive fork-join search, the method balances exploration and exploitation without gradient guidance.

**Strengths:**

- The paper introduces inference-time search to diffusion-based inverse problems with side information, a setting largely unexplored in prior work.
- The proposed reward-tilting formulation allows the use of arbitrary side information (image, text, MRI contrast) without retraining or model modification.

**Weaknesses:**

- The proposed search algorithms lack formal theoretical guarantees for exploration–exploitation optimality or convergence.
- The method relies on heuristic scheduling and requires manual tuning across tasks.
- The computational trade-offs between particle count, reward evaluation cost, and performance are not systematically discussed.
- The experimental comparisons are not fully consistent across baselines. Different methods are evaluated on different task settings or degradation levels, which weakens the fairness and interpretability of the reported improvements.

**Questions:**

- How robust is the search procedure when the side information is noisy, partially mismatched, or misleading?
- The reported DPS baseline results appear significantly weaker than those in prior literature. Could the authors clarify whether this is due to different degradation settings, implementation details, or the influence of side-information conditioning?

---

> ### Author Response · Authors · 2025-11-21
> **Author Response (1/2)**
>
> We thank the reviewer for the insightful comments on the utility and generality of side information in our approach. Based on the reviewers’ suggestions, **we have now included additional experimental results and discussions in Appendix B.1 and B.5**. We hope that these additions will further clarify the novelty and effectiveness of our contributions. Below, we address all the questions and comments raised by the reviewer. We have also updated the **supplementary material** to incorporate these revisions. For the reviewers' convenience, all modifications relative to the original submission are highlighted in blue.
>
> ---
>
> **Q1.**
> "How robust is the search procedure when the side information is noisy, partially mismatched, or misleading?"
>
> **Response:**
> In our revised version, **we have included additional experimental results in Appendix B.5.2** to address this concern, by evaluating how the side information quality affects the performance of our algorithm. We consider \(4\times\) super-resolution using DPS as the baseline sampler for face data. We evaluate the performance of our RFJS algorithm with guidance provided by side information pictures blurred by a Gaussian kernel of varying intensities (1, 3, 5); see Figures 15 and 16 in the revision. We observe that even with degraded (intensity 5) side information, our algorithm (FS: 0.53) improves over the baseline (FS: 1.042) without any side information, and the performance improves as the quality of the side information improves.
>
> ---
>
> **Q2.**
> "The reported DPS baseline results appear significantly weaker than those in prior literature. Could the authors clarify whether this is due to different degradation settings, implementation details, or the influence of side-information conditioning?"
>
> **Response:**
> We believe the reviewer is referring to Table 2 in the main paper. The DPS results there appear weaker because the degradation levels are significantly more severe than those used in prior work. For example, our super-resolution task uses a downsampling factor of 32 (vs. 4 in earlier papers). The motion and Gaussian deblurring tasks use kernels of size 256 (vs. 64 previously), and the noise level is 0.1 instead of 0.05. In the nonlinear and blind deblurring tasks, the noise level is 0.6 (vs. 0.05). Even the box-inpainting masks are larger than those used in earlier studies. In contrast, Table 1 uses the same degradation settings as prior work, and the DPS performance there aligns well with previously reported results.
>
> ---
>
> **Q3.**
> "The proposed search algorithms lack formal theoretical guarantees for exploration–exploitation optimality or convergence."
>
> **Response:**
> Thank you for the question. The focus here is on a principled and practical algorithm with demonstrably superior performance over the SOTA baselines. Theoretical analysis for this problem will be very challenging due to the presence of complicated distributions at intermediate times in the diffusion process. This might require more stylistic modeling of the problem, perhaps as a contextual bandit or POMDP. That is beyond the scope of this algorithm-focused paper. The theoretical analysis is left for future work.
>
> Though we do not have a formal theoretical analysis, we have included an **intuitive and stylized 2D simulation experiment in Appendix B.9** (in the revision), comparing the two search algorithms in a setting where we have both measurements and side information. We use gradient-based guidance for the measurements and search-based guidance for the side information. The results show that our novel RFJ search approach outperforms the Greedy Search method.

---

> ### Author Response · Authors · 2025-11-21
> **Author Response (2/2)**
>
> **Q4.**
> "The method relies on heuristic scheduling and requires manual tuning across tasks."
>
> **Response:**
> We note that the hyperparameters are kept the same across all inverse problem tasks. The only component we tune is the base resampling rate, and even that is selected once per type of side information and then used consistently across all tasks.
>
> ---
>
> **Q5.**
> "The computational trade-offs between particle count, reward evaluation cost, and performance are not systematically discussed."
>
> **Response:**
> Please note that we have discussed these systematically in **Sections B.7 and B.8 of the Appendix** (in the revision). Section B.7 analyzes how the number of particles affects performance, and Section B.8 provides detailed wall-clock comparisons across algorithms. Together, these sections give a clear picture of the trade-offs between particle count, reward-evaluation cost, and overall performance.
>
> ---
>
> **Q6.**
> "Experimental comparisons are not fully consistent across baselines. Different methods are evaluated on different task settings or degradation levels, which weakens the fairness and interpretability of the reported improvements."
>
> **Response:**
> Our main goal is to demonstrate that our method can improve any diffusion-based inverse problem solver in a plug-and-play fashion. The degradation levels differ across baselines because each baseline has a different performance range for different tasks. This variation, however, does not affect the overall conclusion: regardless of the baseline or task difficulty, incorporating side information through our method consistently boosts performance.

---

### Official Review · Reviewer_8H1q · 2025-11-01

**Soundness:** 2
**Presentation:** 3
**Contribution:** 2
**Rating:** 4
**Confidence:** 4

**Summary:**

The authors propose to solve inverse problems by using side information in diffusion model based inverse problem samplers. This could be useful for cases where the observed degradation signal itself might not provide useful information for generating high quality reconstructions and thus using side information could help in these scenarios. In addition to the reward gradient guidance, the authors also propose search methods for non-differentiable rewards. Empirical results are illustrated on the FFHQ dataset for different inverse problems like Super-resoution, non-linear deblur etc.

**Strengths:**

1. The flow of the paper is straightforward to understand although some claims in the main text need more clarification (see weaknesses below)

2. The problem setup seems relevant in the context of diffusion inverse problem solvers and could be useful for a lot of other applications.

**Weaknesses:**

**Re. Theoretical assumptions:**

1. The authors propose to use the reward model r(x_0, s). However, the right distribution to sample from would be the following tilted distribution p(x_0|s,y) \propto p(y|x_0)p(s|x_0)p(x_0). Do the authors assume that p(x_0|s, y) \approx p(x_0|s) which would imply that the side information completely explains the ground truth observation. This is a very strong assumption and might not hold in practice. Can the authors clarify more on this aspect?

2. What is the intuition behind setting \eta=0 in Eq. 6. This implies that the authors assume E[x_0|x_t, y] \approx E[x_0|x_t] which is again a strong assumption? Is this primarily for computational convenience? I see that the authors have some theoretical results which validate these assumptions to some extent for a small t but the claims are still seem dubious here. I would request the authors to clarify this aspect in more details in the main text.

**Re. Missing related work**:  The authors highlight some work under Reward-gradient guidance in Section 2 and highlight that recent works are typically used for semantic generation tasks rather than inverse problems (line 126). However this claim is incorrect and I think some related work is missing. For instance the idea of using reward maximization with KL regularization for test time inference is not new and has been explored in [1] from the perspective of optimal control and in [2] from the perspective of variational inference. Both works explore these ideas in the context of inverse problems.

[1] Variational Control for Guidance in Diffusion Models - Pandey et al.

[2] Divide-and-conquer posterior sampling for denoising diffusion priors - Janatai et al.

**Re Empirical Comparisons:**

1. Given the large body of work on solving inverse problems, the paper lacks empirical comparisons with state of the art methods which do not require any side information. For instance comparisons are made against DPS which is outdated. This is important since if the method cant outperform solvers which dont require any side information for the same input degradation, its practical utility is very limited.

2. While the reconstructions in Fig. 3 look decent and the differences between the proposed method and baseline DPS look noticeable, I wonder how cherry picked these samples are? This is because in terms of quantitative results, the reconstruction gains (in terms of PSNR and SSIM) and perceptual gains (LPIPS) are marginally better than the baselines. I can see that the main differences are in terms of the FaceSimilarity metric but Im curious why the other metrics are only marginally better.

3. Can the authors report runtime estimates in Table 1 too? This would help in a holistic comparison between different baselines and the proposed method in terms of the improvement in different metrics vs the additional compute required.

**Questions:**

See weaknesses above

---

> ### Author Response · Authors · 2025-11-21
> **Author Response (1/2)**
>
> We thank the reviewer for the helpful comments regarding the clarity of our presentation and the broader relevance of our work. Based on the reviewers’ suggestions, **we have now included additional experimental results and discussions in Appendix B.1 and B.5**. We hope that these additions will further clarify the novelty and effectiveness of our contributions. Below, we address all the questions and comments raised by the reviewer. We have also updated the **supplementary material** to incorporate these revisions. For the reviewers' convenience, all modifications relative to the original submission are highlighted in blue.
>
> ---
>
> **Q1.**
> "Do the authors assume that $p(x_0 \mid s, y) \approx p(x_0 \mid s)$ which would imply that the side information completely explains the ground truth observation?"
>
> **Response:**
> We clarify that we **do not** make this assumption. In fact, our formulation explicitly ensures that $p_{0\mid Y, S}(x_0 \mid y, s) \neq p_{0\mid S}(x_0 \mid s)$ and our approach in Proposition 1 is specifically designed to *sample from the true posterior*,
> $$
> p_{0\mid Y, S}(x_0 \mid y, s) \propto p_0(x_0)\, p_{Y\mid 0}(y \mid x_0)\, p_{S\mid 0}(s \mid x_0),
> $$
> as correctly restated by the reviewer. A possible confusion may have arisen from our use of the reward function $r(x_0; s)$ to model $p_{0\mid S}(x_0 \mid s)$ rather than $p_{0\mid Y, S}(x_0 \mid y, s)$. This modeling choice is deliberate, as it allows us to isolate the intrinsic relationship between $x_0$ and $s$ while decoupling the influence of $y$. Importantly, through the derivations in Proposition 1, this construction still effectively samples from $p_{0\mid Y, S}(x_0 \mid y, s)$.
>
> ---
>
> **Q2.**
> "What is the intuition behind setting $\eta = 0$ in Eq. 6? … I would request the authors to clarify this [approximation] aspect in more details."
>
> **Response:**
> We provide the intuition, experimental evidence, and theoretical justification for this approximation.
>
> *Intuition.*
> The measurement $y = A x_0 + \sigma_y z$ is a noisy observation of $x_0$, while $x_t$ represents a noisy version of $x_0$ at diffusion time $t$. When $t$ is small, $x_t$ retains a high correlation with $x_0$ and therefore contains more information about $x_0$ than the noisy measurement $y$. In this regime, the additional conditioning on $y$ contributes negligibly, leading to $E[x_0 \mid y, x_t] \approx E[x_0 \mid x_t]$.
>
> *Empirical evidence.*
> We evaluated Eq. (6) with $\eta = 0$ and $\eta = 1$, and observed no measurable difference in performance, confirming that including or omitting the $y$-dependent correction term has negligible effect.
>
> | Task             | η   | FS (↓) | PSNR (↑) | LPIPS (↓) | SSIM (↑) |
> |------------------|-----|--------|----------|------------|-----------|
> | **Inpainting**   | 0.0 | 0.292  | 28.39    | 0.1365     | 0.857     |
> |                  | 1.0 | 0.288  | 28.47    | 0.1356     | 0.857     |
> | **Super-Resolution** | 0.0 | 0.388  | 25.21    | 0.2274     | 0.694     |
> |                  | 1.0 | 0.379  | 25.25    | 0.2257     | 0.695     |
> | **Gaussian-Deblur**  | 0.0 | 0.331  | 26.22    | 0.1981     | 0.712     |
> |                  | 1.0 | 0.334  | 26.15    | 0.1993     | 0.709     |
> | **Motion-Deblur**    | 0.0 | 0.342  | 26.62    | 0.1911     | 0.737     |
> |                  | 1.0 | 0.334  | 26.55    | 0.1921     | 0.734     |
>
>
>
> *Theoretical justification.*
> While computing $E[x_0 \mid y, x_t]$ exactly is intractable, a tractable first-order approximation is
> $$
> E[x_0 \mid y, x_t] \approx E[x_0 \mid x_t] - \left(\frac{1-\alpha_t}{\alpha_t}\right)\eta \, \nabla_{x_t}\|y - A E[x_0 \mid x_t]\|_2^2.
> $$
> As $t \to 0$, we have $\alpha_t \to 1$, and therefore $(1-\alpha_t)/\alpha_t \to 0$. This ensures that the correction term vanishes regardless of $\eta$, leading to
> $$
> E[x_0 \mid y, x_t] = E[x_0 \mid x_t] + o(1),
> $$
> where the residual error is provably small. Thus both theory and empirical results support setting $\eta = 0$.

---

> ### Author Response · Authors · 2025-11-21
> **Author Response (2/2)**
>
> **Q3.**
> "I think some related work is missing."
>
> **Response:**
> Thank you for the pointer to [1, 2]. We will include a discussion of these references in our revised manuscript. Below, we give a short comparison between our work and each of these papers.
>
> The focus of [1] is fundamentally different from ours, as it does not incorporate side information into the inverse-problem formulation. We model the relationship between the data $x$ and the side information $s$ using a reward $r(x, s)$-tilted posterior, whereas [1] models the reward to maximize the likelihood of the data $x$ with respect to the measurement $y$. Furthermore, [1] assumes the reward is *differentiable*, while we make no such assumption. Our inference-time search framework supports non-differentiable rewards.
>
> The formulation and algorithm proposed in [2] also do not incorporate side information. Instead, [2] uses distribution tilting aligned with the measurement $y$ using Langevin dynamics, and is more similar to DAPS. It is **not** an **inference-time search** algorithm and is therefore very different from our approach.
>
> ---
>
> **Q4.**
> "… the paper lacks empirical comparisons with state-of-the-art methods which do not require any side information … Comparisons are made against DPS which is outdated."
>
> **Response:**
> Our original submission already included comparisons with MPGD and DAPS, as mentioned in lines 407–410. Due to page limits, these comparisons appear in the Appendix—please see **Appendix B.3 and B.4**.
>
> We also emphasize that our method performs significantly better than these baselines, as demonstrated in the original experiments. To further alleviate concerns, **we have now included additional experimental results and discussions in Appendix B.1 and B.5**. The qualitative results appear in Figures 13 and 14, and the quantitative results appear in Tables 6 and 7. These experiments further support that our algorithm provides improved (identity-preserving) reconstructions even in severely ill-posed settings.
>
> ---
>
> **Q5.**
> "While the reconstructions in Fig. 3 look decent … the main differences are in terms of the FaceSimilarity metric but I'm curious why the other metrics are only marginally better."
>
> **Response:**
> The experimental results already show that our method improves over the baselines, but classical metrics like PSNR/LPIPS/SSIM often fail to capture perceptual or identity-level similarity. Much more significant improvements appear in the task-specific perceptual metrics such as **Face Similarity** and **CLIP Score**. Figure 7 in our original submission shows qualitatively better reconstructions than DPS, yet PSNR/LPIPS/SSIM fail to reflect these perceptual improvements. We had previously included this observation in Appendix B.1, Figure 7.
>
> We have now created **a new expanded subsection (Appendix B.1)** with extensive results further illustrating this mismatch. Figures 7, 8, Table 3, and the large-scale results in Figures 13–14 (Appendix B.5.1) all demonstrate how our method achieves substantial perceptual improvements even when classical metrics show only marginal gains.
>
> ---
>
> **Q6.**
> "Can the authors report runtime estimates in Table 1 too? This would help in a holistic comparison between different baselines and the proposed method."
>
> **Response:**
> Due to space constraints, we report detailed wall-clock comparisons in **Appendix B.8**. Since this is an inference-time search algorithm, it is natural to trade off compute for performance. Appendix B.7 analyzes the performance scaling with particle count. We will include runtime summaries in the revised version of Table 1.

---

### Official Review · Reviewer_nJoA · 2025-11-01

**Soundness:** 3
**Presentation:** 2
**Contribution:** 2
**Rating:** 4
**Confidence:** 4

**Summary:**

This author proposes a reward-model based method to aid the inverse problem solving with side information with diffusion models such as reference image, text and so on.

**Strengths:**

1. This reward function to utilize side information is a novel contribution.
2. The injection of side information seems to improve performance

**Weaknesses:**

1. The side information is explored in prior works such as [1], [2], [3]. Authors should extensively discuss with those methods
2. Even though reward function may not be differentiable, why not using DPO to fine tune the model. How is the fine-tuned performance compared with inference-time search? Is the inference-time searching adding more computational cost.
3. The experiments do not show the advantage of this method significantly. Inverse problem solvers already achieve quite good performance in most common cases. I expect this method should be more effective in challenging scenarios, for example 32x super-resolution, inpainting with heavy noise, or heavy blur etc. I also expect your method largely outperform DPS if implemented correctly.
4. The code is not available and I cannot validate without the code.


[1] CLAY: A Controllable Large-scale Generative Model for Creating High-quality 3D Assets

[2] Generative Diffusion Prior for Unified Image Restoration and Enhancement

[3] Prompt-tuning Latent Diffusion Models for Inverse Problems

**Questions:**

Please show the experimental results on hard inverse problems, compare with [1],[2],[3], and at least provide some pseudo-code, and then I will consider improving my rating.

---

> ### Author Response · Authors · 2025-11-21
> **Author Response (1/2)**
>
> We thank the reviewer for the positive comments about the novelty and effectiveness of our approach. Based on the reviewers’ suggestions, **we have now included additional experimental results and discussions in Appendix B.1 and B.5**. We hope that these additions will further clarify the novelty and effectiveness of our contributions. Below, we address all the questions and comments raised by the reviewer. We have also updated the **supplementary material** to incorporate these revisions. For the reviewers' convenience, all modifications relative to the original submission are highlighted in blue.
>
> ---
>
> ### **Q1.**
> “Please show the experimental results on hard inverse problems, compare with [1],[2],[3], and at least provide some pseudo-code, and then I will consider improving my rating.”
>
> **Response:**
> Thank you! We have now addressed each of these points. Please see our detailed response below.
>
> ---
>
> ### **Q2.**
> “The experiments do not show the advantage of this method significantly … I expect this method should be more effective in challenging scenarios, for example 32x super-resolution, inpainting with heavy noise, or heavy blur etc.”
>
> **Response:**
> We address this question in two parts.
>
>
> #### **Part I. Perceptual metrics**
>
> We would like to emphasize that the experimental results included in our submission already show the superior performance of our method compared to the baselines. We believe that a possible confusion may have arisen from comparing only the traditional metrics such as PSNR, LPIPS, and SSIM. While our method shows improvement in these metrics, much more significant improvements can be seen in the target task–specific perceptual metrics, namely Face Similarity and Clip Score.
>
> For example, Fig. 7 in our original submission demonstrates that our reconstruction is qualitatively better than that of DPS. We observed that even in such cases, classical metrics such as PSNR/LPIPS/SSIM often fail to capture identity/perceptual similarity because they primarily reflect low-level pixel agreement. Thus, the improved perceptual quality of our method does not necessarily translate into proportional improvements in these metrics.
>
> We had included this observation in Figure 7 of the submitted manuscript. We have now **included a new subsection, Appendix B.1, with additional experimental results**, which discusses this observation in detail. In particular, we include reconstructed and ground-truth images where the superior performance of our method is visually obvious, yet PSNR/LPIPS values do not adequately reflect the perceptual gains. Please see Figures 7, 8, and Table 3 in the Appendix.
>
>
>
> #### **Part II. Hard inverse problems**
>
> The reviewer is indeed correct that our method significantly outperforms the baselines in hard inverse problems. To further illustrate this, **we have now conducted the following additional experiments:** (i) super-resolution at **12×, 16×, 32×**, (ii) box inpainting: **96×96**, **116×116**, (iii) motion deblur with kernel size **256×256**, and (iv) Gaussian deblur with kernel size **81×81**. The qualitative results are shown in Figures 13 and 14, and the quantitative results are shown in Tables 6 and 7, **included in a new subsection, Appendix B.5.1**. These extensive experiments further reinforce our central claim that the proposed method yields consistently higher-quality reconstructions across all evaluated tasks. This is particularly evident in the substantial gains observed in Face Similarity.
>
> ---
>
> ### **Q3.**
> “The code is not available and I cannot validate without the code.”
>
> **Response:**
> Please note that the pseudo-code is given in Algorithm 1 in Appendix A.3. We now provide the complete reproducible code. We have also included the link to the repo in the abstract of the revised manuscript. For convenience, you can check the code [here](https://anonymous.4open.science/r/sideinfo-search-reconstruction-4A32/README.md).

---

> ### Author Response · Authors · 2025-11-21
> **Author Response (2/2)**
>
> ### **Q4.**
> “The side information is explored in prior works such as [1], [2], [3]. Authors should extensively discuss with those methods.”
>
> **Response:**
> Thank you for the pointers to these papers. Please note that the original manuscript already contains a discussion about “Inverse problems with side information” (lines 114–122). We will include a discussion of these references in the revised manuscript. Below, we provide a brief comparison between our work and each of these papers.
>
> **CLAY [1]** is a conditional generative model for 3D asset synthesis that uses text, image views, and simple 3D cues as conditioning signals, injecting them into a native 3D diffusion model through trained modality-specific modules (e.g., cross-attention adapters). This approach enables controllable generation, but it is *not* an inverse-problem method, as it does not operate under a measurement model. By contrast, our work focuses specifically on diffusion-based inverse problems and incorporates side information in a training-free, modality-agnostic way via a reward that guides inference-time search.
>
> Although **[2]** uses the term “side information,’’ it actually considers multiple measurements of the same ground truth, where the relationship between the measurements and the clean image is either known or can be estimated parametrically. In contrast, our side information may come from *different modalities* (text, reference images, MRI contrasts), with an unknown and nonlinear relationship to the target image. Because this relationship is not expressible as a forward model, we instead incorporate it implicitly via a reward guiding inference-time search, allowing our method to handle arbitrary modalities without retraining.
>
> We also note that **[3]** is already cited in our paper. Its focus is fundamentally different, as it does not incorporate side information. Instead, it optimizes the latent code and text-prompt embedding during inference to better exploit the prior. In contrast, our framework explicitly uses external side information through reward-guided search that reshapes the inference distribution.
>
> ---
>
> ### **Q5.**
> “Why not use DPO to fine-tune the model… Is the inference-time searching adding more computational cost?”
>
> **Response:**
> As noted in Section 3.2 (lines 4–10), one natural approach to incorporating side information is to use training-based methods such as DPO-style fine-tuning. However, these approaches require fine-tuning the diffusion model to accept the side information as input, requiring large amounts of paired data and substantial computational resources. Moreover, this conditioning must be retrained separately for each modality (text, images, MRI contrasts), making it increasingly complex and inflexible.
>
> In contrast, our method can leverage *arbitrary* side information at inference time, even if the pretrained diffusion model was never trained for that modality, because side information appears only through a reward guiding inference-time search.
>
> As our experiments show, inference-time search with \(N\) particles produces higher quality output but increases computational cost compared to the \(N=1\) vanilla setting. This cost–quality relationship is analyzed in Appendices B.7 and B.8, where we show that increasing the number of particles improves reconstruction quality at the expense of increased runtime. With only 4–8 particles, we obtain significant improvements, as reported in the main results.

---

### Meta-Review · Area_Chair_4n5u · 2026-01-24

**Summary:**

Main concerns from reviewers include:
1. Novelty and Discussion of Related Works. Reviewer nJoA noted that "side information is explored in prior works," while Reviewer 8H1q identified missing papers on test-time inference and reward maximization for inverse problems (e.g., Variational Control and Divide-and-conquer).
2. Experiment and baselines. Many reviewers are skeptical of the reported results. . Reviewer 8H1q questioned if the qualitative results were "cherry picked" given that traditional quantitative metrics like PSNR and SSIM were only "marginally better than the baselines." Furthermore, Reviewer nJoA and Reviewer 1jpL pointed out that the DPS baseline results appeared "significantly weaker" than in prior literature.
3. Concerns regarding theoretical justifications (Reviewer 8H1q and Reviewer 1jpL).

**Reviewer Concerns:**

The authors updated the papers by providing some experiments in appendix, mostly focusing on providing additional results on harder inverse tasks, justifying why some reported metrics failed to capture the perceptual comparison between two methods, and provide many discussion with related works and clarification for the theoretical justification.

The scope of the paper or the novelty of the paper can remain a concern given the lack of discussion of related test-time scaling papers in diffusion models, or it requires better writing to surface the contribution. While additional experiments were provided, most of these results were provided in few number of qualitative examples, where the performance of the proposed method is similar to the baseline in many cases. The author further argue the limitation of the evaluation metrics, making it more difficult to evaluation the performance gain provided by the method. The additional clarification for related works and theories could help alleviate some of the reviewers' concerns.

**Reviewer Scores:**

Reviewer nJoA: the key concerns regarding experiments and novelty.The additional experiments can partially address the reviewer's request for harder experiments. The reviewer might still find the relative advantage of the proposed methods compare to the baselines, including some alternative methods like DPO, to be small given the limited qualitative results. I will guess that the reviewer has a slight chance to increase the score from borderline reject to more positive.

Reviewer 8H1q:  The reviewers concerned about missing discussion of related works, theoretical justification, and validity of the experiments. While concerns about related works can be alleviated if the reviewers buy the clarification, It's hard to imagine that the concern about potential cherry-picking can be fully addressed. As a result, the reviewer might keep the same score.

Reviewer 1jpL: the reviewer concerns about the lack of formal theoretical guarantee, inconsistent baselines, and computational trade-offs. I think the computational trade-off is addressed, while the theoretical guarantee or concerns about validity of baselines are only partially addressed. I think 1jpL might at best keep the score if not decrease it looking at other reviewers.

Reviewer kQJ7: requested about additional experimental results and baselines inconsistency. I think the concerns regarding the experiments are partially addressed by the additional results in the appendix, while the concerns about various approximation can remain. The reviewer might keep the score.

---

### Decision · Program_Chairs · 2026-01-26

Reject